# VGR: VISUAL GROUNDED REASONING

**Jiacong Wang**[1,2*]  **Zijian Kang**[2*]  **Haochen Wang**[1,2*]  **Haiyong Jiang**[1‡]  **Jiawen Li**[2]
**Bohong Wu**[2]  **Ya Wang**[2]  **Jiao Ran**[2]  **Xiao Liang**[2†]  **Chao Feng**[2‡]  **Jun Xiao**[1‡]

[1]School of Artificial Intelligence, University of Chinese Academy of Sciences
[2]ByteDance Inc.
`{haiyong.jiang, xiaojun}@ucas.ac.cn`
`chaofeng.zz@bytedance.com`     `wangjiacong20@mails.ucas.ac.cn`

## ABSTRACT

In the field of multimodal chain-of-thought (CoT) reasoning, existing approaches predominantly rely on reasoning on pure linguistic space, which inherently suffers from language bias and is largely confined to math or science domains. This narrow focus limits their ability to handle complex visual reasoning tasks that demand comprehensive understanding of image details. To address these limitations, this paper introduces VGR, a novel reasoning multimodal large language model (MLLM) that can replay the visual memory during thinking just like humans. Unlike traditional MLLMs, VGR first thinks the question and detects relevant regions that may help solve problems, then, the visual memory from the critical area is extracted to assist reasoning. To achieve this, we curate a large-scale SFT dataset called VGR-SFT that contains reasoning data with mixed vision grounding and language deduction. This teaches VGR to think and actively choose grounding areas for key information before answering, and we propose a dynamic visual memory replay stage to integrates the corresponding information into the reasoning process, enhancing multimodel comprehension. Experiments on the LLaVA-NeXT-7B baseline show that VGR achieves superior performance on multimodal benchmarks requiring comprehensive image detail understanding. Compared to the baseline, VGR uses only 30% of the image token count while delivering scores of +4.1 on MMStar, +7.1 on AI2D, and +12.9 improvement on ChartQA. The data is available at `https://huggingface.co/BytedanceDouyinContent/VGR`.

## 1 INTRODUCTION

Large language models (LLMs) have demonstrated remarkable reasoning capabilities, particularly in complex problem-solving scenarios such as mathematical deduction and scientific analysis. Systems like OpenAI-o1 (OpenAI, 2024) and DeepSeek-R1 (Guo et al., 2025a) exemplify this progress, achieving state-of-the-art performance on benchmarks requiring logical inference and algorithmic thinking, where the crux seems to be large-scale Reinforcement Learning (RL) (Sutton et al., 1998; Wang et al., 2025c; Hao et al., 2025) with verifiable rewards (Shao et al., 2024b).

Recent advancements in multimodal reasoning have sought to extend these capabilities to vision-language tasks, often by distilling knowledge from powerful LLMs into multimodal architectures (Huang et al., 2025; Dong et al., 2025; Yang et al., 2025; Wang et al., 2025b; Dong et al., 2024; Ren et al., 2024; 2025). While promising results have emerged in math and science domains, studies consistently reveal a critical limitation: language bias (Jiang et al., 2025; Wang et al., 2024d; Xu et al., 2024), *i.e.*, over-reliance on linguistic priors leads to systematic performance drops in perception-heavy tasks.

To address this limitation, we propose **V**isual **G**rounded **R**easoning (VGR). Instead of reasoning solely in the linguistic space, we argue that models should perform targeted visual analysis during reasoning to identify key regions of interest that are directly relevant to the question. VGR extends the conventional text-only chain-of-thought to a multimodal reasoning trace, *allowing the model to*

---

*∗ Equal Contribution.  † Project Lead.  ‡ Corresponding Author.

*selectively retrieve visual memory on demand*, thereby enhancing the accuracy and interpretability of multimodal reasoning. This approach mirrors human cognition: we reason not only through language but also by recalling and mentally simulating visual content.

Specifically, we design a novel *self-driven* visual memory replay module to retrieve and replay the visual memory. The selective replay is controlled by the model via a predefined special signal: when the model requires visual grounding during reasoning, it generates a replay signal, prompting VGR to fetch corresponding visual tokens from the visual memory to augment its reasoning process. We implement VGR with a novel architecture, the visual memory is constructed from visual representations of high-resolution crops, and a pooling strategy is adopted to further enhance efficiency, the whole design makes the model even more efficient than conventional MLLMs.

To learn this format of instruction, we construct a new visual grounded reasoning dataset embedded with visual clues, marking the attempt to explicitly model visual region attention in multimodal reasoning. Unlike prior works that either rely on text-only chains-of-thought (Xu et al., 2024; Wang et al., 2024d) for multimodal tasks or enforce rigid multi-turn interactions (Wu & Xie, 2024; Shao et al., 2024a; Qi et al., 2024; Xiao et al., 2024), our dataset empowers models to autonomously attend to arbitrary visual regions during reasoning. Notably, all grounding areas in the dataset are voluntarily generated by the model itself, avoiding manual annotation bias. To construct this dataset, we first use an existing model to generate a cold-start dataset, which is then refined via a rejection sampling pipeline and further expanded using annotations from a custom-trained annotation model.

We conduct extensive experiments on this novel framework, results under fair comparison show that our method outperforms the baseline on multiple datasets, such as +6.4 on the MMStar (Chen et al., 2024) and +14.1 on ChartQA (Masry et al., 2022), while utilizing only 0.3× visual tokens. These findings not only underscore the effectiveness of visual memory replay in improving visual-linguistic reasoning, but also establish a new paradigm for enhancing computational efficiency in MLLMs.

In summary, our contributions are threefold:

- We introduce VGR, a new visual reasoning framework for MLLM, which enables the model to dynamically attend to visual content during inference, enhancing reasoning accuracy with fine-grained visual details.
- We build the visual grounded reasoning data with visual cues, the dataset empowers models to freely attend to arbitrary visual memory during reasoning, contrasting with prior works relying on text only chains of thought or rigid interactions.
- Extensive experiments on VGR demonstrate that our model outperforms the LLaVA-NeXT baseline in downstream tasks while using only 0.3× the number of image tokens. As the quantity of image tokens increases, this performance gap becomes even more pronounced.

## 2   RELATED WORKS

**Multimodal Large Language Models.** Pioneering MLLM frameworks like Flamingo (Alayrac et al., 2022) and BLIP-2 (Li et al., 2023a) establish foundational architectures for cross-modal understanding using cross-attention. Subsequently, LLaVA (Liu et al., 2023c) emerged as a more efficient, scalable, and modular framework, combining a vision encoder with a large language model through a simple linear projection layer. Its instruction-tuning paradigm demonstrated competitive performance, emphasizing the power of aligned vision-language supervision. Building on this idea, recent advancements (Liu et al., 2024a; Wang et al., 2024c; Bai et al., 2025; Wang et al., 2024a; 2025a; Lu et al., 2024; Wu et al., 2024; Zhu et al., 2025; Lei et al., 2025; Wang et al., 2024b; Gu et al., 2024; 2025) push the boundaries of efficiency, scalability, and task complexity. For instance, Qwen2.5-VL (Bai et al., 2025) integrates dynamic resolution, and InternVL3 (Zhu et al., 2025) emphasizes the importance of larger-scale native multimodal pretraining. These models serve as strong baselines for a variety of real-world applications.

**Reasoning MLLMs.** The groundbreaking success of advanced reasoning LLMs like OpenAI-o1 (OpenAI, 2024) and DeepSeek-R1 (Guo et al., 2025a) has inspired efforts to extend such capabilities into multimodal domains. Prior attempts (Wang et al., 2024d; Jiang et al., 2025) found that CoT prompting even brings performance degradation for perception-heavy tasks due to the accumulation of language bias. Therefore, current approaches mainly focus on incentivizing MLLMs

to solve difficult math and science problems with image inputs. For instance, Vision-R1 (Huang et al., 2025) first leverages MLLMs to generate detailed captions for provided images and then queries DeepSeek-R1 (Guo et al., 2025a) to obtain a dataset for cold initialization. Other approaches, such as VLM-R1 (Shen et al., 2025) and Visual-RFT (Liu et al., 2025), directly adopt GRPO (Shao et al., 2024b) for open-ended visual grounding, where RL consistently outperforms SFT. In parallel, another line of work enhances high resolution grounding via supervised CoT training: Zoomeye (Shen et al., 2024) leverages human-like zooming with tree-based exploration, while Chain-of-Spot (Liu et al., 2024c) employs interactive reasoning for spot based visual search. These methods yield notable gains on benchmarks such as HR-Bench (Wang et al., 2025d) and V* Bench (Wu & Xie, 2024), highlighting the complementary role of CoT style supervision for perception intensive multimodal reasoning tasks. This paper presents an alternative methodological approach that complements existing perspectives. Our framework seeks to incentivize the "grounding-then-answering" capability in MLLMs, requiring the model to systematically develop two critical competencies: (1) frequent autonomous selection of task-relevant image regions through deliberate focus mechanisms, and (2) contextualized reasoning based on these visually grounded observations.

## 3 VISUAL GROUNDED REASONING

In this section, we elaborate on the framework and model of VGR, in Figure 1. To unlock the visual grounding reasoning capabilities, we introduce a novel visual memory replay mechanism, allowing the model to attend to arbitrary image regions by retrieving corresponding image tokens during the reasoning on the fly.

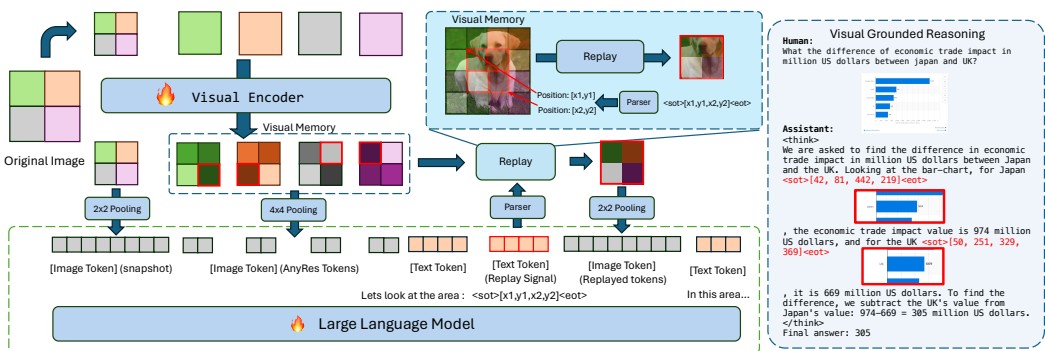

Figure 1: Overview framework of our method. On the left side of the figure, we apply the AnyRes strategy to the original image while maintaining a visual memory pool that stores detailed visual features. When a visual memory replay signal is detected, VGR retrieves image tokens from this visual memory pool, thereby enriching the visual clues available for reasoning. On the right side of the figure, we present an example of VGR in operation: it enables the MLLM to inspect key regions on demand.

The dynamic visual memory replay module of VGR retrieves image tokens generated by the vision encoder and adapter. Leveraging LLaVA's AnyRes approach for high resolution image encoding, we first resize the input image to dimensions $H \times W$ where $H$ and $W$ are divisible by $p = 336$. The resized image $\mathbf{P} \in \mathbb{R}^{H \times W \times 3}$ is then partitioned into non-overlapping $p \times p$ patches:

$$\mathbf{P}_{ij} = \mathbf{P}[p * i : p * (i+1), p * j : p * (j+1)]. \tag{1}$$

The corresponding image tokens are processed by the vision-encoder and adapter, yielding token embeddings in the language space:

$$\mathbf{T}_{i,j} = \mathcal{F}_{adapter}(\mathcal{F}_{vision}(\mathbf{P}_{ij})) \in \mathbb{R}^{\frac{p}{s} \times \frac{p}{s} \times c}, \tag{2}$$

where $s$ denotes the size of the vision patch and $c$ denotes the channel number of latent features. Like in LLaVA, the image tokens from each crop are flattened to a 1D sequence and fed in the LLM separately. We further concatenate the feature of each patch representation to a unified image feature $\mathbf{S} \in \mathbb{R}^{\frac{H}{s} \times \frac{W}{s} \times c}$ for later use, which serves as the visual memory.

The dynamic visual memory replay mechanism relies on fine grained visual feature for retrieval, to preserve high resolution visual details while maintaining training and inference efficiency, we propose an expand-then-compress strategy. Specifically, we scale up the maximum crop count of LLaVA's AnyRes approach from 4 to 16 patches and introduce a vision feature compression layer using 2D pooling. To balance resolution and computational cost, we adopt $2 \times 2$ pooling for snapshot compression and $4 \times 4$ pooling for high resolution AnyRes token compression empirically.

Compared to the baseline, which employs maximum 2,880 tokens per image (576 tokens per shot, including 1 snapshot image and 4 AnyRes crops), VGR achieves superior efficiency. Our method uses only 144 tokens for the snapshot and a maximum of 720 tokens for high resolution crops, reducing token usage by 70% while expanding supported resolutions by $5\times$. This design guarantees VGR to maintain fine grained visual information for retrieval while lowering computational overhead.

To enable the MLLM selectively attend to specific visual regions, we introduce a replay control signal for the model. Each replay region is defined via a grounding area notation: `<sot>[x1, y1, x2, y2]<eot>`, where `[x1, y1]` denotes the top-left corner and `[x2, y2]` the bottom-right corner of the region. The visual tokens will be retrieved once such signal is detection. The MLLM is encouraged to generate these signals during inference in demand to extend visual clues.

During inference, VGR monitors the model output and, upon detecting signal token `<eot>`, parses the preceding content to extract the region coordinates. If valid, the model retrieves image tokens corresponding to this region from the feature map $\mathbf{S}$ and appends them after the control signal. Specifically, for a region defined by coordinates $(x_1, y_1)$ and $(x_2, y_2)$, the dynamic visual memory replay module extracts the corresponding feature patch $\mathbf{R}_{x_1,y_1,x_2,y_2}$, the extracted feature map $\mathbf{R}_{x_1,y_1,x_2,y_2}$ is then down-sampled with $2 \times 2$ pooling and flattened into a 1D token sequence. They are fed into the LLM immediately following the visual replay signal token.

Implementing supervision for the dynamic visual memory replay is straightforward. We simply add the retrieved image tokens $\mathbf{R}_{x_1,y_1,x_2,y_2}$ to the training sequence after the replay signal and optimize the model with the standard supervised fine-tuning. The signal tokens as well as text tokens are supervised with cross-entropy loss, while all image tokens (both from the original input and the replay regions) are excluded from the loss computation. To further enhance the model's region selection capability, we introduce an auxiliary detection loss that encourages accurate area predictions.

The detection loss is important because coordinates for retrieval are actually represented as numbers, $\mathcal{L}_{\text{det}}$ operates as a straightforward regression task to precisely align spatial locations, since cross-entropy on tokenized boxes may struggle with quantization errors and discontinuous predictions. Therefore, combining both allows the model to leverage continuous regression for accurate localization. Specifically, the detection loss is a combination of $\ell_1$ loss and GIoU loss:

$$\mathcal{L}_{\text{det}} = \ell_1 + \beta \ell_{\text{GIoU}}, \tag{3}$$

where $\ell_1$ Loss measures the absolute difference between predicted bounding box coordinates and ground truth. We set $\beta = 2$ following common practices. For a bounding box parameterized by center coordinates $(x_c, y_c)$, width $w$, and height $h$, the formula is:

$$\ell_1 = |\hat{x}_c - x_c| + |\hat{y}_c - y_c| + |\hat{w} - w| + |\hat{h} - h|, \tag{4}$$

where $\hat{x}_c, \hat{y}_c, \hat{w}, \hat{h}$ are predictions. GIoU loss is computed by:

$$\ell_{\text{GIoU}} = 1 - \left( \frac{\text{InterArea}}{\text{UnionArea}} - \frac{C - \text{UnionArea}}{C} \right), \tag{5}$$

where $C$ is the smallest box enclosing both predicted and ground truth boxes:

$$C = (x_C^2 - x_C^1) \cdot (y_C^2 - y_C^1), \tag{6}$$

where $x_C^1 = \min(x_1, \hat{x}_1), x_C^2 = \max(x_2, \hat{x}_2), y_C^1 = \min(y_1, \hat{y}_1), y_C^2 = \max(y_2, \hat{y}_2)$. The detection head we utilized is a small MLP that maps the hidden states of `<eot>` to a 4-dimensional box.

## 4 VISUAL REASONING DATA CURATION

VGR learns to the visual reasoning through our reasoning data with replay signal an visual memory, with the proposed three stage data construction pipeline as shown in Figure 2. The cold start data is generated with an existing large instruction model and further refined with reject sampling. Then, we train an annotation model to annotate data from more domains.

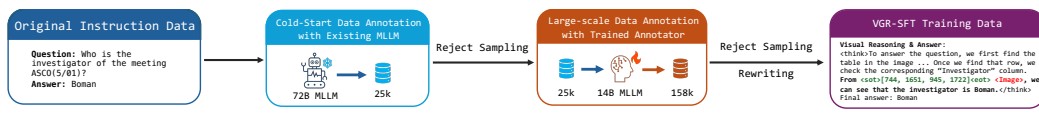

Figure 2: Overview framework of `VGR` data pipeline. We use an existing large MLLM to annotate initial cold-start data, and train a smaller annotator model to scale up the amount of training data. A reject sampling and refinement pipeline is adopted to improve the data quality.

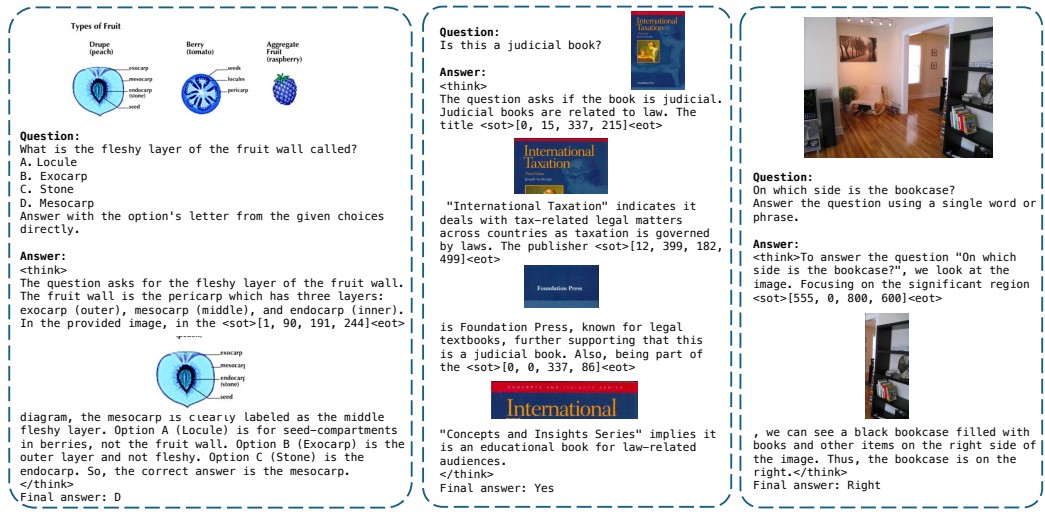

Figure 3: Example of training data in **VGR-SFT**.

## 4.1 COLD-START WITH INSTRUCTION MODEL

The initial instruction data with replay capabilities is generated using an existing vision language model. Specifically, given an image and a corresponding question, the model is prompted to generate both a reasoning chain and an answer. Concurrently, we require the model to localize all key regions in the image relevant to the answer and explicitly reference these regions before describing their content. These key regions are designated as replayed areas during training. We adopt a 72B huge MLLM (Bai et al., 2025) as the cold-start model, due to its exceptional instruction following capabilities, output diversity, and strong performance in both object detection and visual reasoning tasks. To standardize the annotation format, we prompt the model to encode detection results in JSON, which includes bounding boxes and semantic labels for each key region.

## 4.2 REJECT SAMPLING

Following the recent advances in RL (Guo et al., 2025a), we propose a similar reject sampling pipeline for valid data selection. First, we employ **Format Verification** to ensure answer parseability. This involves two checks: (1) verifying that answers can be extracted by locating the designated "Final Answer" section; (2) ensuring bounding boxes and labels are formatted in valid JSON. Next, **Correctness Verification** assesses the accuracy of answers derived from reasoning chains. For closed ended tasks (e.g., OCR, MCQ), we use ANLS (Average Normalized Levenshtein Similarity) to quantify correctness by comparing generated answers with ground truths. For open ended tasks, we leverage a MLLM to semantically align reasoning chains with reference answers. Incorrect answers are discarded, while inaccurate ones undergo rewriting: the final answer is replaced with ground truth, and reasoning chains for open ended tasks are iteratively revised for coherence. Finally, **Visual Grounding Verification** validates the correctness of visual replay areas. During data preparation, each visual replay area is annotated with a bounding box and semantic label. We crop these areas and use a MLLM to check alignment between cropped content and annotated labels. Additionally, we intentionally expand bounding box areas to encourage the trained model to retain contextual information during reasoning, enhancing its ability to handle complex visual semantic dependencies.

### 4.3 DATA SCALING WITH ANNOTATION MODEL

During the reject sampling, we notice the cold-start data generated by the existing instruction model exhibits a high rejection rate and slow generation speed. To address these limitations, we train an annotation model using the cold-start data that passes the reject sampling pipeline. Empirically, we adopt a smaller 14B MLLM (Zhu et al., 2025) as annotation model, we augment it with the cold-start data, we also use reasoning data from the Open-R1 distilled dataset (Face, 2025) to generalize the conventional reasoning ability to our visual reasoning task. With the knowledge and pattern learned from cold-start data, the smaller annotation model significantly improves the pass rate from 14% in cold-start to 40% and also speedup annotation for 3.2× times. This allows us to scale the amount of our training data with low cost.

#### 4.3.1 TRAINING DATA

In the last step, we refine the annotated data that passes through the reject sampling pipeline, a MLLM is used to revise the reasoning chains, enhancing reasoning robustness. The refinement aligns the data with our predefined template while eliminating ambiguous or redundant content. The refined data is subsequently utilized to train the final reasoning model, ensuring its capacity to generate structured and coherent responses. The final training data **VGR-SFT** is curated from LLaVA's official training data, which aligns strictly with the baseline in fair data comparison, where the composition of the data is shown in Table 1 and examples are shown in Figure 3. You can find more details in the Appendix D.

Table 1: The number of data generated from each dataset.

| Method | Data Size | Data Type |
|---|---|---|
| AI2D (Kembhavi et al., 2016) | 12.5k | ScienceQA |
| LLaVA-COCO (Liu et al., 2023b) | 12.3k | General VQA |
| GQA (Hudson & Manning, 2019) | 39.2k | General VQA |
| ChartQA (Masry et al., 2022) | 11.2k | OCR |
| DVQA (Kafle et al., 2018) | 25.2k | OCR |
| DocVQA (Mathew et al., 2021) | 6.0k | OCR |
| OCRVQA (Mishra et al., 2019) | 51.6k | OCR |
| **Total** | 158.1k | - |

## 5 EXPERIMENTS

### 5.1 EXPERIMENT SETTINGS

We validate VGR on LLaVA-NeXT (Liu et al., 2024b) setting following the recent practices, which is a well-known and fully open-sourced baseline for multimodal training from scratch. The visual encoder is CLIP-ViT-L/14@336 (Radford et al., 2021) and the base LLM is Vicuna-v1.5 series (Chiang et al., 2023), including 7B and 13B versions. Following (Liu et al., 2024b), VGR has two training procedures: pre-training and supervised fine-tuning. The pre-training data is LLaVA-558K (Liu et al., 2024a), while the fine-tuning data is the combination of LLaVA-NeXT-770K (Liu et al., 2024b) and our self-constructed 158K data. Notably, to ensure a fair comparison, all datasets constructed are derived from the original SFT data of LLaVA-Next without introducing any additional data. The LR for pre-training stage is set to 1e-5 and 2e-5 for fine-tuning stage with Vicuna-7B. We set the learning rate of ViT to 1/10 of the base learning rate follow the LLaVA-NeXT's setting.

### 5.2 COMPARISON WITH EXISTING METHODS

We compare our VGR with a wide range of existing vision-language models on various benchmarks, including MMStar (Chen et al., 2024); ChartQA (Masry et al., 2022); DocVQA (Mathew et al., 2021); TextVQA (Singh et al., 2019); InfoQA (Mathew et al., 2022); AI2D (Kembhavi et al., 2016); RealWorldQA (Grok, 2024); POPE (Li et al., 2023b). For clarity, we note that Sample represents the image token compression or downsampling scheme used, while Vtoken indicates the maximum number of image patch tokens. The top results are highlighted in **bold**. All results are drawn either

Table 2: Comparison with existing vision-language models on various vision-language benchmarks.

| Method | DownSample | #Vtoken | LLM | MMS* | Chart | Doc | Text | Info | AI2D | RWQA | POPE |
|---|---|---|---|---|---|---|---|---|---|---|---|
| Qwen-VL-Chat-7B (Bai et al., 2023) | Cross-Attn | 1024 | Qwen-7B | 34.5 | 66.3 | 62.6 | 61.5 | - | 57.7 | 49.3 | 74.9 |
| Visual CoT (Shao et al., 2024a) | No | 576 | Vicuna-7B | - | 22.8 | 49.3 | **66.9** | - | - | - | 86.5 |
| DeepSeek-VL-7B (Lu et al., 2024) | Conv2D | 576 | DeepSeek-7B | 40.5 | 59.1 | - | 64.9 | - | 65.3 | 54.2 | 85.6 |
| LLaVA-v1.5-7B (Liu et al., 2023a) | No | 576 | Vicuna-7B | 33.1 | 18.2 | 28.1 | 46.1 | 25.8 | 54.8 | 54.8 | 85.9 |
| LLaVA-NeXT-7B (Liu et al., 2024b) | No | 2880 | Vicuna-7B | 37.6 | 54.8 | 77.4 | 64.9 | 37.1 | 66.6 | 57.8 | 86.5 |
| LLaVA-NeXT-7B† | 2×2 \| 4×4 | 864 | Vicuna-7B | 37.2 | 58.7 | 70.2 | 60.5 | 34.7 | 68.5 | 56.8 | 87.8 |
| **VGR-7B** | 2×2 \| 4×4 | 864 | Vicuna-7B | 41.7 | 67.7 | 73.7 | 63.9 | 39.8 | **73.7** | **59.8** | **88.2** |
| **VGR-7B** | 2×2 \| 2×2 | 3024 | Vicuna-7B | **43.6** | **72.8** | **79.9** | 65.9 | **42.9** | 73.4 | 59.5 | 87.8 |

Table 3: **Ablations on different backbone and high resolution benchmark.** Abbreviations in the table correspond to the following models: Qwen2.5 refers to Qwen2.5-7B-Instruct, siglip to SigLIP-SO400M/14@384, InternViT to InternViT-300M/14@448-v2.5, CLIP-ViT to CLIP-ViT-L/14@336, and Vicuna to Vicuna-7B-v1.5.

| Training Settings | V* Bench | HR-Bench8K | MMStar | ChartQA | TextVQA | RWQA | AI2D |
|---|---|---|---|---|---|---|---|
| Qwen2.5+SigLIP | 55.5 | 44.9 | 51.6 | 64.0 | 62.0 | 75.6 | 63.3 |
| Qwen2.5+SigLIP+VGR | 67.5↑ 12.0 | 56.1↑ 11.2 | 54.1↑ 2.5 | 74.2↑ 10.2 | 65.5↑ 3.5 | 77.9↑ 2.3 | 65.4↑ 2.1 |
| Qwen2.5+InternViT | 56.0 | 45.2 | 51.4 | 75.2 | 68.2 | 76.0 | 61.7 |
| Qwen2.5+InternViT+VGR | 69.8↑ 13.8 | 58.3↑ 13.1 | 55.2↑ 3.8 | 78.1↑ 2.9 | 71.7↑ 3.5 | 79.4↑ 3.4 | 64.9↑ 3.2 |
| Vicuna+CLIP | 56.4 | 41.1 | 37.2 | 58.7 | 60.5 | 68.5 | 56.8 |
| Vicuna+CLIP+VGR | 67.7 ↑ 11.3 | 52.9↑ 11.8 | 43.6 ↑ 6.4 | 72.8 ↑ 14.1 | 65.9↑ 5.4 | 73.4↑ 4.9 | 59.5↑ 2.7 |

from the original papers or from the official reproduction results reported by LMMs-Eval (Zhang et al., 2024), whereas our results are consistently obtained using LMMs-Eval. In particular, † denotes our reproduction setting with a maximum of 20 local images using LLaVA-NeXT (Liu et al., 2024b) and visual memory feature pooling (2×2 for the base crop and 4×4 for local crops), with replay visual memory features further processed by 2×2 pooling.

As shown in Table 2, our VGR consistently outperforms strong alternatives, including Qwen-VL-Chat (Bai et al., 2023), Visual CoT (Shao et al., 2024a), DeepSeek-VL-7B (Lu et al., 2024), LLaVA-v1.5-7B (Liu et al., 2023a), and LLaVA-NeXT-7B (Liu et al., 2024b). In particular, it achieves the best results on benchmarks requiring fine-grained comprehension of high-resolution images. Moreover, when taking into account the average number of visual tokens, VGR delivers superior performance with $0.3 \times$ visual tokens compared to the original LLaVA-NeXT, suggesting that focusing the model on *specific regions* is substantially more effective than merely increasing the number of visual tokens. As the number of image tokens further increases, this performance gap becomes even more pronounced. The comparison between VGR and the Zoomeye method in terms of performance and inference cost is available in Appendix A.2.

## 5.3 Ablation Studies

**Ablations on more backbones and dataset.** In Table 3, we present experiments designed to further verify the generalizability of the VGR. We replaced key model modules (including the visual encoder/ViT backbone and base language model/LLM) while strictly upholding experimental fairness: we used the same dataset throughout (with no additional data introduced) and only adjusted the LLM and ViT backbones. In the table, the last section (configured with Vicuna-7B-v1.5 and CLIP-ViT-L/14@336) aligns with the original architecture of LLaVA-NeXT-7B (serving as a baseline). The experimental results confirm that our VGR framework not only delivers consistent performance improvements across diverse visual encoders, base LLMs, but also improve fine grained high resolution benchmarks (V* Bench (Wu & Xie, 2024) and HR-Bench 8K (Wang et al., 2025d)) that need visual grounding capabilities.

**Ablations on different data formulations.** The ablation study in Table 4 demonstrates that the visual reasoning capacity requires *both* grounding boxes and reasoning. When either component is removed, whether by eliminating visual memory (w/o Memory) or disabling the reasoning process (w/o Reasoning), performance consistently degrades. This suggests that while each component makes a partial contribution, the two components can complement each other to achieve the best.

Table 4: **Ablations on different data formulations.** [†] indicates our reproduction on same setting with pooling. "w/o Memory" indicates that only the reasoning process is preserved without grounding and replay. "w/o Reasoning" means we remove the reasoning process.

| Method | MMStar | ChartQA | DocVQA | TextVQA | InfoVQA | AI2D | RWQA | POPE |
|---|---|---|---|---|---|---|---|---|
| LLaVA-NeXT-7B[†] | 37.2 | 58.7 | 70.2 | 60.5 | 34.7 | 68.5 | 56.8 | 87.8 |
| **VGR-7B** | **41.7** | **67.7** | **73.7** | **63.9** | **39.8** | **73.7** | 59.8 | **88.2** |
| w/o Memory | 39.7 | 66.2 | 73.2 | 63.0 | 39.3 | 72.7 | **60.6** | 87.5 |
| w/o Reasoning | 39.3 | 59.6 | 72.5 | 61.9 | 38.5 | 72.8 | 59.3 | 87.8 |

Table 5: **Ablations on public available CoT data.** We evaluate different dataset on our setting (indicated by [†]), an extra post-training stage is added for LLaVA-CoT and MMPR following their recommendations.

| Data | SFT | Post-Train | MMStar | ChartQA | DocVQA | ScienceQA |
|---|---|---|---|---|---|---|
| LLaVA-NeXT[†] | 770K | – | 37.2 | 58.7 | 70.2 | 70.3 |
| LLaVA-CoT[†] | 770K | 100K | 39.6 | 58.8 | 64.4 | 76.5 |
| MMPR[†] | 770K | 660K | 40.7 | 55.1 | 68.3 | **82.1** |
| **VGR-7B** | 770K + 158K | – | **41.7** | **67.7** | **73.7** | 70.4 |

**Compare with public available CoT data.** We compare the effectiveness of our reasoning data (which explicitly utilizes regions of interest) against vanilla reasoning datasets such as LLaVA-CoT (Xu et al., 2024) and MMPR (Wang et al., 2024d). As shown in Table 5, the model with region-of-interest guidance focuses more on relevant visual areas, leading to improved overall performance. In contrast, direct adoption of complex reasoning datasets like (Xu et al., 2024; Wang et al., 2024d) yields results even worse than the baseline. This may stem from the accumulation of language bias during multimodal reasoning, highlighting that appropriately integrating visual features of regions of interest significantly aids accurate inference.

**Ablations on detection loss.** In Table 6a, we study the effectiveness of the auxiliary detection loss. Since boxes are represented by floating-point coordinates normalized to the [0, 1] range. $\mathcal{L}_{det}$ operates as a straightforward regression task to precisely align spatial locations, since cross entropy on tokenized boxes may struggle with quantization errors and discontinuous predictions. Therefore, combining both allows the model to leverage continuous regression for accurate localization.

**Ablations on dynamic visual memory replay.** In Table 6b, we systematically evaluate the efficacy of dynamic visual memory replay through ablation studies. Results show that excluding dynamic visual memory replay where the model merely outputs regions of interest *without* incorporating corresponding image features into the LLM input sequence leads to significantly limited performance improvements. This highlights the critical gain from integrating image features of boundary regions into the reasoning process, as it enables the model to leverage fine grained visual details for more accurate predictions.

**Ablations on different reasoning data type.** In Table 6c, we analyze the performance differences across reasoning data of varying types. Using raw data after annotation during supervised fine-tuning introduces longer contexts, but this also makes the model prone to make mistakes, which does not benefit general question answering. In contrast, after summarizing and condensing them into relatively shorter rewritten data, the data and reasoning process are less confusing, therefore enabling the model to develop stronger grounded reasoning abilities.

**Ablations on different replayed strategies.** In Table 7, we investigate the trade-off between pooling performance by differentiating the pooling steps for base images, local images, and Replay visual memory, as these components exhibit distinct levels of importance. The base image, which encapsulates the most comprehensive understanding of the entire visual content, demands a balance between global context and spatial detail. Local crops, while useful, often contain redundant information due to overlapping regions, justifying coarser pooling. Dynamic visual memory replay, however, represent specific regions of interest (RoIs) critical to task-solving and thus require finer-grained feature preservation compared to standard local crops. Empirically, the optimal configuration employs 2×2 pooling for both base and Replay visual memory to retain critical details, while applying

Table 6: **Ablations on each component**, including (a) the introduction of detection loss, (b) whether to apply the dynamic visual memory replay after predicting bounding boxes, and (c) the type of reasoning data. By default, we enable detection loss and dynamic visual memory replay with a maximum of 20 local crops.

| (a) Detection loss. | | | | (b) Visual memory replay. | | | (c) Reasoning Data Type. | | | |
|---|---|---|---|---|---|---|---|---|---|---|
| $\mathcal{L}_{det}$ | MMStar | ChartQA | DocVQA | Replay | MMStar | ChartQA | Type | MMStar | ChartQA | AI2D |
| – | 39.8 | 65.5 | 72.8 | – | 39.7 | 66.2 | Annotated | 40.7 | 64.5 | 71.7 |
| ✓ | **41.7** | **67.7** | **73.7** | ✓ | **41.7** | **67.7** | Rewritten | **41.7** | **67.7** | **73.7** |

Table 7: **Ablations on different replayed strategies.** Different pooling strides for each type of image are important. We utilize different pooling size for the final image feature and **Replayed** image feature. We also use visual memory to reduce the costs of training and inference.

| Base | Local | Replayed | #Crops | #Memory | #Vtoken | MMStar | ChartQA | DocVQA | TextVQA | InfoQA |
|---|---|---|---|---|---|---|---|---|---|---|
| – | – | – | 4 | No | 2880 | 37.2 | 58.7 | 70.2 | 60.5 | 34.7 |
| 2×2 | 4×4 | 2×2 | 4 | Yes | 288+20 | 37.5 | 53.4 | 52.0 | 57.0 | 30.1 |
| 2×2 | 4×4 | 2×2 | 20 | Yes | 864+100 | 41.7 | 67.7 | 73.7 | 63.9 | 39.8 |
| 2×2 | 4×4 | 2×2 | 20 | No | 864+360 | 41.2 | 65.9 | 74.0 | 63.6 | 40.1 |
| 2×2 | 2×2 | 2×2 | 20 | Yes | 3024+100 | **43.6** | **72.8** | **79.9** | **65.9** | **42.9** |

4×4 pooling to local images to mitigate redundancy without significant information loss. In rows 3 and 4, we present the experimental results comparing two approaches for obtaining replay visual features: one using visual memory and the other cropping images via bounding boxes before re-inputting them into the ViT. The results demonstrate that our use of visual memory not only achieves comparable performance but also reduces training and inference costs by utilizing fewer image tokens.

## 5.4 Test-Time Replay Token Scaling.

To further improve the performance, we investigate the possibility of test-time replay token scaling. Specifically, we adopt a larger image cropping scheme during testing to generate more image tokens while keeping the pooling strategy unchanged. The results shown in Table 8 indicate that a further scaling of tokens is also helpful, and this phenomenon is especially prominent in OCR-related tasks. We set 64 as the maximum number of cropped images, in practice, most images in the fine-tuning data do not reach such resolution level.

Table 8: **Test Time Image Tokens Scaling.** We apply a larger image resolution cropping scheme during testing to obtain more image tokens. Other setting is same as the Table 7.

| Base | Local | Replayed | #Crops | #Vtoken | MMStar | ChartQA | DocVQA | TextVQA | InfoQA |
|---|---|---|---|---|---|---|---|---|---|
| 2×2 | 4×4 | 2×2 | 20 | 864+100 | 41.7 | 67.7 | 73.7 | **63.9** | 39.8 |
| 2×2 | 4×4 | 2×2 | 64 | 2592+400 | **42.9** | **67.9** | **76.3** | **63.9** | **42.9** |

## 6 Conclusion

In this work, we propose VGR for enhanced multimodal comprehension. VGR enables MLLMs to reason on visual clues and selectively attend to informative regions on demand. To achieve this, we introduce a selective feature replay module, which allows the model to focus on crucial regions, thereby enhancing fine-grained comprehension—particularly for small regions in high-resolution inputs. We also curate a large-scale reasoning dataset, VGR-SFT, which for the first time integrates visual information into dense reasoning tasks. Extensive experiments on VGR demonstrate considerable improvements across multiple benchmarks, validating the effectiveness of our approach.

**Discussion.** Our method has limitations that warrant future research. First, VGR is currently implemented on LLaVA Liu et al. (2023b) architecture, exploring stronger visual encoders and LLMs could further enhance performance. Another avenue is integrating reinforcement learning (RL), a more generalized and diverse reasoning process may be achievable with RL.

## ACKNOWLEDGMENTS

This work is supported by the National Natural Science Foundation of China (62476262, 62271467, 62306297,62306296), the Beijing Nova Program, Beijing Natural Science Foundation (4242053, L242096), the Postdoctoral Fellowship Program of CPSF (GZB20250411), and the Fundamental Research Funds for the Central Universities.

## ETHICS STATEMENT

Our research is grounded in ethical practices, with particular attention paid to the responsible use of data. All datasets employed in this study are publicly available and well-established within the computer vision community. Specifically, our benchmarking was conducted on LLaVA (Liu et al., 2023b). Our use of this data is in accordance with their provided licenses and intended academic purpose.

## REPRODUCIBILITY STATEMENT

We are committed to ensuring the reproducibility of the research presented in this paper. To this end, comprehensive implementation details for our models and experiments are provided in Appendix, including the training procedures and all hyperparameters used. Furthermore, upon acceptance of this paper, all source code, datasets, and trained model checkpoints will be made publicly available.

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

APPENDIX

## A    MORE ABLATION EXPERIMENTS ANALYSIS OF **VGR** IN THE MAIN TEXT

In Table 2 of the main text, the experiment incorporating the complete components of VGR achieved the optimal results. We also list the outcomes when either component is removed—by eliminating grounding cues ("w/o Grounding") or disabling the reasoning process ("w/o Reasoning"). Comparing the results without grounding (using only reasoning data with visual memory replay) against the baseline LLaVA-NeXT-7B (first row) demonstrates that even without grounding, the performance exceeds the original baseline, highlighting the rationality and critical role of our reasoning data construction. Additionally, comparing the results without reasoning (retaining grounding with visual memory replay) against the LLaVA-NeXT-7B baseline shows that these outcomes also surpass the original baseline, validating the effectiveness and significance of using grounding boxes for image visual memory replay.

In Table 6a of the main text, the two rows of experimental results ablate the impact of adding detection loss. The results without detection loss still outperform the LLaVA-NeXT-7B baseline, confirming the validity of the other two components: visual memory replay and reasoning data. Table 6b presents ablation results for visual memory replay; removing visual memory replay still yields performance superior to the baseline, underscoring the rationale behind the detection loss and reasoning data components. Table 6c ablates reasoning data, and the results without it still exceed the baseline, demonstrating the effectiveness of detection loss and visual memory replay.

**More trainging details.** As stated in Section 5 of our main text, we follow the exact hyperparameter settings of LLaVA-NeXT, training all data for one epoch. The learning rate is 1e-5 for the pre-training stage and 2e-5 for fine-tuning (Vicuna-7B). The ViT learning rate is set to one-tenth of the base rate. We use a warmup ratio of 0.03, a cosine scheduler, and zero weight decay.

### A.1    PERFORMANCE COMPARISON OF VGR-SFT.

As shown in Table 9, we first observe that data scaling is critical for VGR: VGR-SFT trained on 158K data clearly outperforms models (Cold-Start/Annotator-Data) trained on 25K data across multiple benchmarks. However, cold-start data scalability is limited by slow annotation speed and a high reject rate. To address this, we trained a smaller-size annotator model (14B MLLM), where cold-start data training eases format and instruction complexity, making it feasible for smaller models to perform annotation. We selected the 14B MLLM for its strong benchmark performance and compatibility with our training framework, and results confirm its effectiveness: compared to the cold-start baseline (Qwen2.5-VL-72B), our annotator-based models are 9x more efficient (3.2x faster inference, 2.9x higher accuracy, from 14% to 40% pass rate) while enabling us to scale more data with limited resources.

For fair comparison, we used 25K data (from two sources) for VGR training: "VGR Cold Start 25K" refers to data annotated by Qwen2.5-VL-72B, while "Annotator Data 25K" denotes data annotated by the 14B MLLM. The two datasets achieved similar VGR performance, demonstrating the effectiveness of our scaling strategy with a smaller annotator model. We will add the full data scaling experiment to the manuscript.

Table 9: **Performance Comparison of VGR-SFT.**

| Model | MMStar | ChartQA | DocVQA | TextVQA | InfoQA | AI2D |
|---|---|---|---|---|---|---|
| LLaVA-NeXT-7B | 37.2 | 58.7 | 70.2 | 60.5 | 34.7 | 68.5 |
| **VGR** Cold Start 25K | 37.4 | 61.2 | 72.6 | 62.3 | 37.1 | 70.4 |
| **VGR** Annotator Data 25K | 37.7 | 60.9 | 72.8 | 61.9 | 36.8 | 70.5 |
| **VGR** 158K | **41.7** | **67.7** | **73.7** | **63.9** | **39.8** | **73.7** |

### A.2    COMPARISON WITH ZOOMEYE.

In Table 10, we present a detailed comparison between our VGR framework and ZoomEye (Shen et al., 2024), along with key methodological distinctions and experimental results. ZoomEye adopts

a recursive tree based search paradigm to simulate human like "zooming" behavior: it first splits images into hierarchical sub-patches and conducts reasoning through iterative node exploration. Though this training free design is innovative and effective for visual grounding, it incurs substantial computational overhead repeated patch splitting and tree traversal lead to slow reasoning speed, especially for high resolution images that demand deep recursion. By contrast, our VGR leverages a selective visual memory replay mechanism, which directly retrieves task-relevant visual tokens from a preconstructed feature pool, thus eliminating the need for recursive search entirely.

Quantitatively, the averaged wall clock time per question for ZoomEye reaches 48.46s on V* Bench and 55.52s on HR-Bench 8K. In comparison, our VGR achieves significantly faster inference, with average per-question times of only 7.5s (V* Bench) and 9.2s (HR-Bench 8K), highlighting the efficiency advantage of our feature retrieval-based design over ZoomEye's recursive search approach.

Table 10: **Comparison with ZoomEye: Inference Time and Performance on High-Resolution Benchmarks.**

| Model | V* Bench | Time | HR Bench | Time |
|---|---|---|---|---|
| LLaVA-NeXT-7B | 56.4 | 1.04s | 41.1 | 1.11s |
| LLaVA-NeXT-7B w/ Zoom Eye | 71.7 | 48.5s | 44.4 | 55.5s |
| **VGR** 7B | 67.7 | 7.51 s | 52.9 | 9.23 s |

# B MORE EXPERIMENTS OF **VGR** ON DIFFERENT MLLMS

## B.1 ABLATIONS ON DIFFERENT POOLING STRATEGIES.

Table 11: **Ablations on different pooling strategies.** Different pooling strides for each type of image are important. We utilize 2×2 for **Base** image feature and **Replayed** image feature, and 4×4 for **Local** images feature. It is noted that above the horizontal line are the results for the original data without VGR-SFT data.

| Model | Base | Local | Replayed | Crops | Vtoken | MMStar | ChartQA | DocVQA | TextVQA | InfoQA |
|---|---|---|---|---|---|---|---|---|---|---|
| LLaVA-Vicuna7B | – | – | – | 4 | 2880 | 37.6 | 54.8 | 77.4 | 64.9 | 37.1 |
| LLaVA-Vicuna13B | – | – | – | 4 | 2880 | 40.4 | 62.2 | 77.5 | 66.9 | 41.3 |
| LLaVA-Vicuna7B | 2×2 | 4×4 | – | 4 | 288 | 37.5 | 53.4 | 52.0 | 57.0 | 30.1 |
| LLaVA-Vicuna7B | 2×2 | 4×4 | – | 20 | 864 | 41.3 | 60.2 | 71.7 | 62.7 | 38.4 |
| LLaVA-Qwen2-7B | 2×2 | 4×4 | – | 20 | 864 | 39.4 | 49.8 | 78.2 | 58.5 | 39.4 |
| *With VGR-SFT data and replay image feature* | | | | | | | | | | |
| LLaVA-Vicuna7B | 2×2 | 4×4 | 2×2 | 20 | 864+100 | 41.7 | 67.7 | 73.7 | 63.9 | 39.8 |
| LLaVA-Vicuna7B | 2×2 | 2×2 | 2×2 | 20 | 3024+100 | 41.7 | 67.7 | 73.7 | 63.9 | 39.8 |
| LLaVA-Vicuna13B | 2×2 | 2×2 | 2×2 | 20 | 3024+100 | 44.6 | 71.7 | 78.6 | 64.9 | 41.8 |
| LLaVA-Qwen2-7B | 2×2 | 2×2 | 2×2 | 20 | 3024+100 | 46.9 | 62.7 | 82.5 | 61.9 | 42.5 |

To illustrate the model performance under different settings, we correct a typo and incorporate additional experimental results in Table 11, including replacing Vicuna-7B with Qwen2-7B on the LLaVA-NeXT 13B and LLaVA-NeXT baselines. We sincerely apologize for the typo in the main text, where the results of the first two rows were incorrectly stated as the raw baseline and VGR setting.

## B.2 HOW INACCURATE BOUNDING BOX REPLAY IMPACTS REASONING.

To directly examine how inaccurate bounding box replay impacts visual grounded reasoning, we evaluate two inference settings: (1) attaching a random box image tensor while keeping the text box token correct; (2) replacing both the text box token and the image tensor with a random box. The corresponding results are shown in Table 12.

We observe substantial performance degradation in both cases. Incorrect visual features cause large negative effects even when the text token is correct, and performance drops further when both text

and visual features are incorrect. This demonstrates that accurate replay predictions play a crucial role in reasoning. It's noted that in the first line, it does not mean that both the box token and the corresponding image feature are random. Instead, it means that we first randomly crop an image tensor with a box. At the same time, replace the text token of the correct box in the current output ID with the text token corresponding to this box.

Table 12: **Performance Comparison of inaccurate bounding box.**

| Model | Box Text Token | Replay Token | V* Bench | DocVQA | MMStar | ChartQA | TextVQA | AI2D |
|---|---|---|---|---|---|---|---|---|
| **VGR-7B** | random | random | 37.4 | 61.2 | 72.6 | 62.3 | 37.1 | 70.4 |
| **VGR-7B** | right | random | 37.7 | 60.9 | 72.8 | 61.9 | 36.8 | 70.5 |

### B.3 THE RELATIONSHIP BETWEEN REPLAY AND GROUNDING

Using lmms-eval, we evaluate LLaVA-Next and VGR on Referring Expression Comprehension(REC) of RefCOCO, RefCOCOg, and RefCOCO+, reporting both the standard IoU>0.5 accuracy and the IoU metric. Results are shown in Table 13. The results show consistent improvements over the baseline, suggesting that VGR training and data also enhance grounding performance.

Table 13: **Performance Comparison of REC tasks. Where the metric of A means acc@0.5, the metric I means the IoU.**

| Model | RefCOCO_A | RefCOCO_I | RefCOCOg_A | RefCOCOg_I | RefCOCO+_A | RefCOCO+_I |
|---|---|---|---|---|---|---|
| **LLaVA-NeXT-7B** | 0.85 | 0.72 | 0.82 | 0.69 | 0.77 | 0.65 |
| **VGR-7B** | 0.88 | 0.74 | 0.84 | 0.71 | 0.78 | 0.68 |

### B.4 CONDUCT VGR ON OTHER BACKBONES.

VGR is a unified framework involving both architectural design and data construction, so we require a baseline that can be fully trained from scratch with a reasonable implementation cost. LLaVA-Next-7B fits this requirement well. In contrast, the training data for InternVL3 and Qwen2.5-VL is not publicly available. Since VGR requires both pre-training and SFT, a complete reproduction of these data-closed models is not feasible. Nevertheless, as reported in Table 3 of the main paper, we have already evaluated VGR on comparable baselines such as InternViT and Qwen2.5 LLM, where we still observe substantial improvements.

To further address some conern about generalization, we conducted an additional analysis experiment. We applied a post-training stage on InternVL3-8B using LLaVA-Next-770K data and obtained results comparable to those reported in its paper (although these data were already used during its internal training). As a comparison, we then applied the VGR style post-training stage to InternVL3-8B under the same setting. Note that this experiment includes only post-training rather than the full multi-stage VGR pipeline in our paper, and is therefore an analysis-oriented probe experiment. The results are shown in Table 14. Even without the full multi-stage pipeline, VGR-style post-training yields improvements on key benchmarks (DocVQA, ChartQA, TextVQA), indicating that the proposed strategy is transferable across different architectures.

Table 14: **Performance Comparison of serve VGR only in post train stage on InternVL3-8B.**

| Data Training | Inference Style | DocVQA | MMStar | ChartQA | TextVQA | AI2D |
|---|---|---|---|---|---|---|
| LLaVA-NeXT-770K | SFT | 91.6 | 65.2 | 86.4 | 80.9 | 83.0 |
| LLaVA-NeXT-770K+VGR-SFT | VGR | 92.8 | 63.9 | 87.8 | 81.5 | 85.3 |

### B.5 INFERENCE BEHAVIOR AND COST WITH VGR.

The increased latency stems primarily from two factors. First, LLM inference time is largely determined by the next token generation mechanism, making the runtime closely tied to output length. Our baseline LLaVA-NeXT does not perform any reasoning and only generates short answers, allowing it to complete inference quickly. In contrast, models using CoT or thinking-style answers must generate longer reasoning chains, which naturally increases latency. The inference times reported in the paper were measured using H20 GPUs and the Lmms-eval framework. Second, we computed the average number of generated tokens, replay times, run time for VGR-7B on different tasks, as shown in Table 15.

VGR requires visual feature replay and longer outputs during answering, which contributes to the increased inference time. Nonetheless, thanks to rapid developments in the open-source ecosystem—such as vLLM and sglang, a more efficient LLM inference framework is becoming increasingly accessible. We will continue to follow and adopt these technologies to further improve the inference efficiency of our model.

Table 15: **Inference behavior and cost with VGR.**

| Metric | MMStar | V* Bench | ChartQA | DocVQA |
|---|---|---|---|---|
| Repaly Times | 2.0 | 1.7 | 2.4 | 2.1 |
| Generated Tokens | 98 | 110 | 155 | 113 |
| Run Time | 5.84s | 7.51s | 11.57s | 6.51s |

we measured the cases where the model failed to produce coordinate text between the special tokens that could be parsed by our rules, or produced invalid coordinates. Across various downstream benchmarks, the proportion of valid bounding boxes consistently ranges from 97.5% to 98.9%, indicating that only a few corner cases fail after VGR training. Second, we trained a variant of VGR-7B that uses absolute pixel coordinates instead of normalized (0–1) relative coordinates, and compared it with the original setting in Table 16. Benefit from the global template construction enabled by LLaVA's any-resolution design and the visual feature memory mechanism, the absolute-coordinate version performs only slightly worse than the original setting.

Table 16: **Performance Comparison of using absolute pixel coordinates instead of normalized (0–1) relative coordinates.**

| Model | Box Type | DocVQA | MMStar | ChartQA | TextVQA | AI2D | RWQA |
|---|---|---|---|---|---|---|---|
| **VGR-7B** | w/normalized | 73.7 | 41.7 | 67.7 | 63.9 | 73.7 | 59.8 |
| **VGR-7B** | wo/normalized | 71.9 | 40.9 | 64.2 | 62.8 | 70.5 | 56.23 |

## C THE CASE OF VGR

Figure 4 shows a case in VGR-SFT with different formulations, after being processed separately, these three types of data are used to train VGR. The corresponding results are w/o Memory, VGR-7B and w/o Reasoning in Table 6c.

We also present a case study in Figure 6 to illustrate how VGR's design contributes to its success in such scenarios.

## D REASONING DATA PIPELINE

### D.1 DETAILS ON DATA CONSTRUCTION

In this section, we elaborate the details on data curation.

**Reject Sampling.** During the reject sampling, we implement two verification steps with MLLM from online API, which is Doubao1.5-VL (Guo et al., 2025b) in our implementation. We did not

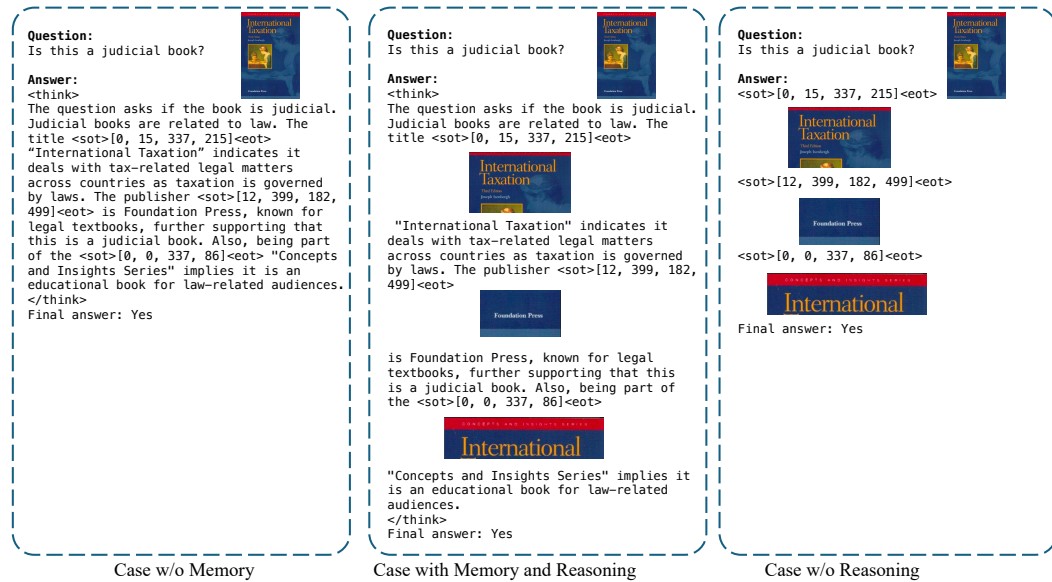

Figure 4: Example of training data in **VGR-SFT** in different formulations.

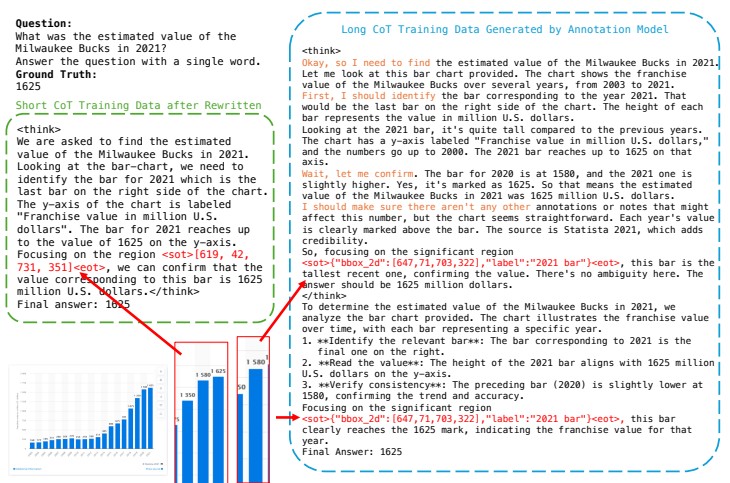

Figure 5: Example generated by our annotation model. We distill core information and the chain-of-thought from long redundant reasoning with reject sampling and rewriting.

adjust the parameters of Doubao1.5-VL and instead used the official default settings, with a maximum response length of 8192. The close-sourced online MLLM is fast and strong, but can not be modified for our specific task and is expensive, therefore we only use this model for well-defined task like checking, filtering and rewriting. The prompts for remote requests are detailed in Table 17, where two distinct prompts are designed for correctness verification of open-ended problems and grounding area verification, respectively. To process responses from the commercial model, we use a simple parser to convert the output into an integer ranging from 0 to 5. A threshold of 3 is applied to filter out noisy data, ensuring the quality of the dataset.

**Data Rewriting.** The data rewriting strategy is introduced to address amendable errors. First, during the reject sampling phase, we perform ground-truth-aligned rewriting to reconcile the generated answers with the ground-truth annotations in our training data. To avoid an absurd change in "Final Answer", we also use the MLLM to align the reasoning chain with final answer. Second, we introduce a format and reasoning process rewriting for all reasoning data, ensuring all data matches the same

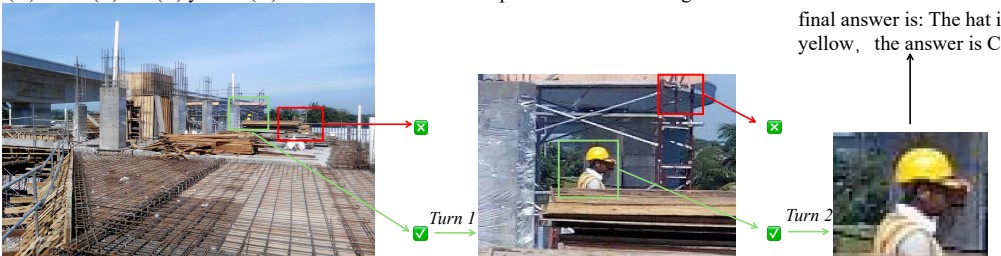

Think step by step and answer the following question, you need to reference the key area with "[x1,x2,y1,y2]" bounding-box format and give the final answer with "Final answer:".
What is the color of the man's helmet?
(A) white (B) red (C) yellow (D) blue     Answer with the option's letter from the given choices.

final answer is: The hat is yellow，the answer is C

Figure 6: Example of how VGR's design contributes to the visual cot.

format, mitigate confusion in the reasoning chains, reduce information leak before the replay and avoid failed answer extraction. The prompts are shown in Table 17

**Annotator Training.**   We train our annotator with two type of data, cold-start data that includes visual reasoning generated by previous steps, and Open-R1 data (Face, 2025) from pure-text reasoning chain distilled from Deepseek-R1 (Guo et al., 2025a). We use the learning rate of $1e-5$, batch size of 128 and trained the model for 6000 steps. During annotator training, we use different prompts for these two types of data, which reduce confusion for the model, the prompt is shown in Table 17. As shown in the next paragraph, we use the combination of these two prompts to generate more diversified answers.

**Annotation Generation.**   We employ distinct prompts to guide our cold-start model and annotation model in generating training data, as outlined in Table 18. For the cold-start model, we provide a highly detailed prompt with an illustrative example to ensure data quality and format consistency. For the annotator, we use a hybrid prompt that integrates visual grounding and Open-R1 prompts, enabling the generation of complex multi-step reasoning akin to DeepSeek-R1's behavior.

## D.2   VISUALIZATION OF DATA AND VGR

To illustrate the effectiveness and necessity of our data pipeline, we show the differences among each segment of our pipeline. The data examples of cold-start, annotated, and the final data are shown in Figure 7. As shown in the figure, our data curation pipeline is able to improve the data quality step-by-step: the annotated data enhances the reasoning complexity and annotation efficiency, while the training data from the rewriting is more concise and easy to learn. To expose more details, data from the annotation model and refined data are illustrated in Figure 5. After training with cold-start data and complex reasoning data from DeepSeek-R1Face (2025); Guo et al. (2025a), the annotation model can generalize its reasoning ability from text-only to visual reasoning. However, this model still easily makes various mistakes, so we still need the reject sampling pipeline and the rewriting module to fix these issues. In this example, the reject sampling and checking module expands the bounding-boxes, aligns them with the correct ground truth, and enriches the context. The rewritten module removes duplicate bounding-boxes, reformats the document, and clarifies the explanations. The rewritten short and clean data is especially valuable for smaller-scale models like Vicuna (Chiang et al., 2023) that only supports 4096 tokens. As shown in Table 6c in the Main Paper, short clean data also performs better than long data in our experiments.

In Figure 8, we visualize the responses of VGR on the MMStar and ChartQA benchmarks trained with our data. VGR automatically and accurately locates target regions in the responses, generates correct reasoning based on the content within these regions, and ultimately provides accurate answers.

## E   FURTHER COMPARISON AND INSIGHTS FROM RELATED WORKS WITH VGR.

We address three distinct novelties of VGR, which differ from previous attempts(including Visual CoT (Shao et al., 2024a), SketchPad (Hu et al., 2024), Refocus (Fu et al., 2025)):

**Native Free-form Visual Reasoning.** VGR enables the VLM to retrieve and reuse visual memory dynamically. Compared with Visual CoT, which trains the model with a multiturn dialog with predefined forms, VGR enables the model to reuse visual memory during complex visual thinking. This allows VGR for flexible "visual re-reading" at any step of the reasoning chain, interleaving summary, reflection, and visual replay—without manual cropping interventions. In contrast, Visual CoT can only retrieve the cropped area once and can only think with a static path.

VGR eliminates the need for a multi-turn dialogue format to guide the model to first output bounding boxes, followed by manual cropping and re-input of relevant visual features and text. Instead, VGR's visual feature replay not only offers greater flexibility in format but also, more importantly, embeds grounding reasoning capabilities within the model itself, enabling the model to spontaneously inspect highly relevant image regions during reasoning, mimicking human-like cognitive processes.

**Full Open Solution with Data and Training.** While methods like SketchPad and Refocus focus on tool-use capacity on advanced commercial MLLMs (such as GPT-4v/o), VGR provides a complete, open-source pipeline including data construction and training. VGR's end-to-end data construction pipeline provides the community with a followable, efficient data iteration framework. VGR also employs joint training of detection loss and autoregressive loss, enhancing the model's grounding capabilities. This allows VGR to enable visual reasoning ability for an open-source VLM with no inherent visual capabilities, while SketchPad and Refocus heavily rely on existing capabilities of closed-source MLLMs.

**Efficient Visual Memory Architecture.** introduces a novel architecture for visual memory. Unlike tool-based methods (such as Visual CoT, SketchPad, and Refocus), that repeatedly process image crops, VGR pre-computes visual tokens once. By using multi-level pooling, it allocates computation to the most critical regions efficiently. As shown in our experiments, this design saves approximately 70% of tokens compared to baseline methods while achieving superior performance.

**CogCom and Chain-of-Focus (CoF), both are great works and worth discussing.** Compared to both of them: VGR proposes a novel architecture with visual memory, which enables visual memory replay during reasoning. Compared to the "grounding then crop" strategy used in CogCom and CoF, the visual memory retrieval operation is more efficient and consistent. VGR pre-computes visual tokens only once, while the crop or zoom-in operations need extra computation on modified views. By using multi-level pooling, VGR further allocates computation to the most critical regions efficiently. As shown in our experiments, this design saves approximately 70% of tokens compared to baseline methods while achieving superior performance.

Compared to CogCom (Qi et al., 2024): CogCom uses predefined operations to retrieve knowledge (e.g., GROUNDING, OCR, CROP_AND_ZOOMING, CALCULATE). However, it does not reason on the retrieved information. On the other hand, VGR learns the memory retrieval operation during reasoning, which is able to analyze, reflect, and plan further operations based on the retrieval result. This makes VGR more flexible for complex quests.

Compared to Chain-of-Focus (Zhang et al., 2025): Chain-of-Focus is a concurrent work with VGR. Compared with CoF, VGR is able to learn from scratch (from an LLM with zero vision prior), which is more challenging, while CoF relies on the existing ability of the pre-trained MLLM. Considering that many backbones trained via VGR have not been well-trained on CoT/reasoning or grounding, VGR makes greater efforts to propose a scalable, high-quality visual reasoning data pipeline. VGR also introduces a detection loss for supervision during training, which effectively improves the grounding and retrieval quality for MLLMs.

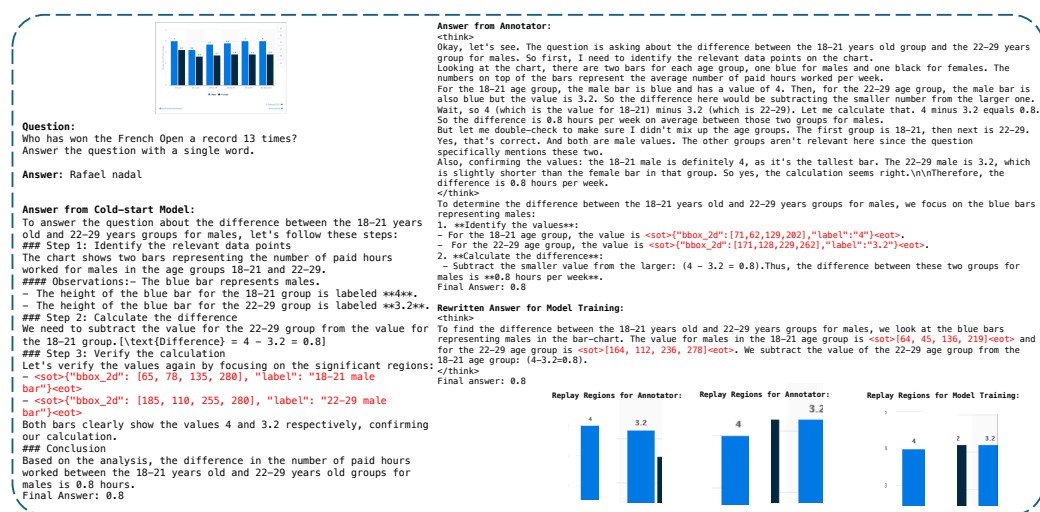

Figure 7: Example of data from original data, cold-start model, annotator and training set.

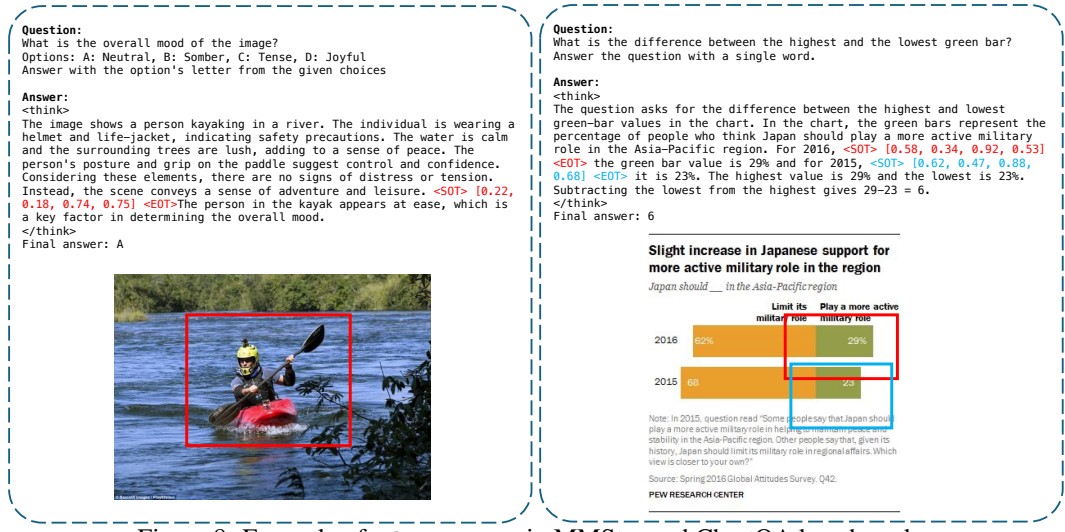

Figure 8: Example of VGR response in MMStar and ChartQA benchmarks.

Table 17: Prompt for **VGR** reject sampling and data rewriting.

| Stage | Prompt |
| --- | --- |
| ***Reject Sampling*** | |
| Correctness Verification | You are an annotator, your goal is to check if the reasoning process is aligned with multimodal question and answer. |
| | You will be given the question, ground truth, the reasoning chain and the original answer. Output an integer from 0 to 5: output 5 if the reasoning chain is aligned with the ground truth (even if the answer has some mistakes), output 0 if the reasoning chain is not aligned with the ground truth. |
| | Question: {question} |
| | Ground truth: {gt} |
| | Reasoning chain: {answer} |
| | Original answer: {final_answer} |
| Visual Grounding Verification | You are an annotator, your goal is to check if the short content description of the bounding box is aligned with the image. I will send you two images: one is the original image and the other is the bounding box area cropped from the original image. |
| | Output a integer from 0 to 5, 0 means the content is not aligned with the content, 5 means well aligned. |
| | Check if the content from the second image is "{content}". |
| ***Data Rewriting*** | |
| Ground-Truth Rewriting | You are an annotator, your goal is to check if the reasoning process is aligned with multimodal question and answer, and rewrite the reasoning chain and the answer to match the ground truth. You can add more details to the answer, but all information introduced by the ground truth should be covered. You will be given the question, ground truth, and the original answer with the reasoning for reference. Output the answer with the reasoning process: think first, then answer the problem. The final answer that matches the ground truth should be written after "Final answer:". |
| | Question:{question} |
| | Ground truth: {gt} |
| | Answer with Reasoning: {answer} |
| Reasoning Chain Rewriting | You are an annotator. Your goal is to check whether the reasoning process aligns with the multimodal question and answer, and rewrite the reasoning chain and the answer to match the ground truth. You can add more details to the answer, but it must cover all the information provided by the ground truth. |
| | You need to remove any redundant, confusing, or incorrect information from the original answer. The rewritten answer should be logical and concise. The answer should follow a strict format: all the thinking parts should be enclosed within <think></think> tags, and then state the ground truth starting with "Final answer:". All location information should be enclosed within <sot><eot> tags; the content of <sot><eot> includes "bbox_2d" and "label", which are simply copied from the original answer and should NOT be changed. You need to reference the area before mentioning any information in the area, and each location should be mentioned only once (i.e., duplicate <sot><eot> tags with the same information should be removed). |
| | You will be provided with the question, the ground truth, and the original answer with its reasoning for reference. Output the answer along with the reasoning process, make the answer fluent, and do not use the ground truth in the reasoning process. You must reference at least one location with <sot>...<eot>, the content of <sot><eot> is copied exactly from the original answer. Think through the problem and the referenced area, and then write the final answer that matches the ground truth after "Final answer:". Only return a single-line final answer, which should strictly conform to the ground truth. |
| | Question: {question} Ground truth: {gt} Answer with Reasoning: {answer} |

Table 18: Prompt for **VGR** model training and data construction.

| Stage | Prompt |
|---|---|
| ***Annotator Training*** | |
| Cold-start Data | Think step by step and answer the following question, you need to reference the key area with <sot>{"bbox_2d":[x1,y1,x2,y2],"label":"..."}<eot> bounding box format and give the final answer with "Final answer:".
The size of the image is {image.width} x {image.height}.
{**original_question**} |
| Open-R1 Data | {**original_question**}
Give step by step reasoning before you answer. This requires engaging in a comprehensive cycle of analysis, summarizing, exploration, reassessment, reflection, backtracing, and iteration to develop well-considered thinking process. You need to use <think> </think> to wrap your reasoning process and answer the final answer enclosed in LaTeX's \boxed tag. |
| ***Data Annotation*** | |
| Cold-start Model | You must locate and focus on the major objects that significantly contribute to solving the question. Prioritize the output of bounding boxes for larger and more significant areas, minimizing the inclusion of smaller, less relevant regions. Output the bounding box coordinates of these key objects in JSON format. As you reason step by step, ensure each step includes detailed considerations such as analyzing the question, summarizing relevant findings, brainstorming new ideas, verifying the accuracy of the current steps, refining any errors, and revisiting previous steps. During this process, emphasize larger and more important areas using the bounding box format <sot>{"bbox_2d":[x1,y1,x2,y2],"label":"..."}<eot> to reference visual details and information. Reference the area before mentioning its content. Finally, answer the question with "Final Answer: xxx". For example:
To answer the question [state the question here], first, we need to identify [describe what needs to be identified], let me focus on this significant region <sot>{"bbox_2d":[x1,y1,x2,y2],"label":"..."}<eot>. You need to replace x1, y1 with the actual pixel coordinates. In this region, I observe [describe what you see in the region, such as the letter xxx]. This observation indicates [explain the significance of what you saw]. Based on this analysis, we can conclude that [continue with the reasoning process]. Therefore, the answer is [state the answer].
Final Answer: [answers]
Now, I will provide you with a Question. Please output the answer with the bounding box incorporated into the reasoning as described above, focusing on larger and key areas, and minimizing small or irrelevant boxes.'
{**original_question**} |
| Annotation Model | Think step by step and answer the following question, you need to reference the key area with <sot>{"bbox_2d":[x1,y1,x2,y2],"label":"..."}<eot> bounding-box format and give the final answer with 'Final answer:'." The size of the image is {image.width} x {image.height}.
{**original_question**}
Give step by step reasoning before you answer. This requires engaging in a comprehensive cycle of analysis, summarizing, exploration, reassessment, reflection, backtracing, and iteration to develop a well-considered thinking process. You need to use <think> and </think> to wrap your reasoning process start and end. During reasoning, reference the key area with {"bbox_2d":[x1,y1,x2,y2],"label":"..."} only at the thinking process. Do not include box information in the final answer. Ensure the final answer appears only once and contains only the solution or conclusion. |
| ***Reasoning Model Training*** | |
| **VGR-SFT** Data | Think step by step and answer the following question, you need to reference the key area with "<sot>[x1,x2,y1,y2]<eot>" bounding-box format and give the final answer with "Final answer:".
{**original_question**} |

