# OpenReview forum: "VGR: Visual Grounded Reasoning"
_ICLR.cc/2026/Conference — ICLR 2026 Poster_

### Official Review · Reviewer_qwzK · 2025-10-31

**Soundness:** 3
**Presentation:** 3
**Contribution:** 3
**Rating:** 6
**Confidence:** 3

**Summary:**

This paper introduces VGR, a multimodal large language model framework that augments chain-of-thought reasoning with selective visual grounding via a dynamic visual memory replay mechanism. VGR detects task-relevant image regions during reasoning, retrieves compressed high-resolution visual features from a maintained visual memory pool, and injects them into the LLM input to improve fine-grained visual-linguistic inference. The experiments on eight visual reasoning benchmarks demonstrate that VGR significantly outperforms existing vanilla, SoTA VLMs.

**Strengths:**

- VGR is a well-motivated and thoughtfully designed approach. The dynamic visual memory replay mechanism is a clever way to enhance the model's ability to focus on relevant image regions during reasoning, which is crucial for fine-grained visual-linguistic tasks.
- The experiments are well-executed and comprehensive, covering a wide range of visual reasoning benchmarks and a series of ablation studies. The results convincingly demonstrate the effectiveness of VGR, with significant performance improvements over existing methods.

**Weaknesses:**

- The novelty of this paper is not clearly articulated, especially in relation to very similar prior work on CogCoM [1] and Chain-of-Focus [2]. Both CogCoM [1] and Chain-of-Focus [2] also focus on enhancing visual reasoning by incorporating dynamic, bounding box-based image retrieval mechanisms. The authors should clearly differentiate VGR from these existing approaches and highlight the unique contributions of their method. The reviewer recognizes L060-061 as an attempt to do so by saying "our dataset empowers models to autonomously attend to arbitrary visual regions during reasoning.", but the same can be said about CogCoM and Chain-of-Focus.
- The experiment section could be strengthened by providing a bit more qualitative analysis into why VGR outperforms other methods. For example, sample a handful of examples (e.g., 10-20) where VGR succeeds but other methods fail, and analyze what aspects of VGR's design contribute to its success in those cases. This would provide deeper insights into the strengths of the proposed approach.

[1] CogCoM: A Visual Language Model with Chain-of-Manipulations Reasoning. https://arxiv.org/abs/2402.04236
[2] Chain-of-Focus: Adaptive Visual Search and Zooming for Multimodal Reasoning via RL. https://arxiv.org/abs/2505.15436

**Questions:**

N/A

---

> ### Author Response · Authors · 2025-11-25
> **Response to Reviewer qwzK**
>
> We are grateful to the Reviewer for your extensive review. We will address your questions point by point below:
> > W1 The novelty with related works.
>
> Thanks for pointing out CogCom and Chain-of-Focus (CoF), both are great works and worth discussing.
>
> Compared to both of them:
> VGR proposes a novel architecture with visual memory, which enables visual memory replay during reasoning. Compared to the "grounding then crop" strategy used in CogCom and CoF, the visual memory retrieval operation is more efficient and consistent. VGR pre-computes visual tokens only once, while the crop or zoom-in operations need extra computation on modified views. By using multi-level pooling, VGR further allocates computation to the most critical regions efficiently. As shown in our experiments, this design saves approximately 70% of tokens compared to baseline methods while achieving superior performance.
>
> Compared to CogCom:
> CogCom uses predefined operations to retrieve knowledge (e.g., GROUNDING, OCR, CROP_AND_ZOOMING, CALCULATE). However, it does not reason on the retrieved information. On the other hand, VGR learns the memory retrieval operation during reasoning, which is able to analyze, reflect, and plan further operations based on the retrieval result. This makes VGR more flexible for complex quests.
>
> Compared to Chain-of-Focus:
> Chain-of-Focus is a concurrent work with VGR. Compared with CoF, VGR is able to learn from scratch (from an LLM with zero vision prior), which is more challenging, while CoF relies on the existing ability of the pre-trained MLLM. Considering that many backbones trained via VGR have not been well-trained on CoT/reasoning or grounding, VGR makes greater efforts to propose a scalable, high-quality visual reasoning data pipeline. VGR also introduces a detection loss for supervision during training, which effectively improves the grounding and retrieval quality for MLLMs.
>
> > W2 More qualitative analysis.
>
> Thank you for the suggestion. We also agree that qualitative case studies would more concretely demonstrate the advantages of VGR. We will include a qualitative analysis in Appendix C.
>
> We will randomly sample 10–20 cases from the various benchmarks used in our experiments, covering chart-based tasks, general visual reasoning datasets, and high-resolution datasets such as HRBench and VStar Bench.
>
> For each case, we will provide:
>
> (a) the original question and the corresponding image,
>
> (b) concise generation traces from both LLaVA-NeXT and VGR (including whether replay was triggered, replay box coordinates, and key reasoning fragments),
>
> (c) a comparative explanation of why VGR succeeds (e.g., replayed high-resolution local features compensate for visual blind spots during linguistic reasoning; memory-based replay avoids repeated ViT encodings and mitigates information loss).
>
> We will further link the observed improvements to the design choices in VGR, such as selective replay, visual memory architecture, and the VGR-SFT data pipeline, to clarify where the performance gains originate.
>
> If you have any further questions or concerns, please feel free to contact us at any time. We are always available and looking forward to further discussions with you. :)
>
> Best regards,
>
> All Authors

---

### Official Review · Reviewer_ysjS · 2025-10-31

**Soundness:** 3
**Presentation:** 3
**Contribution:** 3
**Rating:** 6
**Confidence:** 4

**Summary:**

The paper proposes VGR, a multimodal reasoning framework that lets an MLLM “replay” visual memory on demand during chain-of-thought. It depicts a mechanism when the model emits a special replay signal with a bounding box, VGR fetches the corresponding image tokens from a high-resolution feature map and appends them into the ongoing context for grounded reasoning. The authors also curate VGR-SFT, a supervised dataset mixing reasoning traces with explicit region grounding produced via a cold-start model, rejection sampling, and a smaller “annotator” model for scale. On LLaVA-NeXT-7B backbones, VGR reports strong gains on perception-heavy benchmarks while using ~30% of the visual tokens versus the baseline, with ablations isolating the contributions of replay and a detection loss for box accuracy.

**Strengths:**

1. **Simple, modular mechanism with clear intuition.**
Turning visual inspection into an explicit token-level replay step is elegant and easy to graft onto LLaVA-style stacks. The control token and parser logic are well specified.

2. **Efficiency/performance tradeoff.**
The expand-then-compress feature design + selective replay cuts visual tokens by ~70% yet improves perception tasks

3. **Data contribution.**
 VGR-SFT explicitly couples reasoning with self-proposed RoIs, avoiding manual box bias and encouraging “ground-then-reason” behavior. The pipeline (cold start → reject sampling → annotator) is thoughtful.

**Weaknesses:**

1. **Replay supervision and parser brittleness.**

 The method depends on the model outputting valid coordinates via \<sot\>[x1,y1,x2,y2]\<eot\>; while a detection loss is added, the paper doesn’t quantify (i) the rate of invalid boxes and (ii) sensitivity to coordinate quantization / resolution scaling. Metrics for “replay success rate” and its effect on final accuracy would strengthen the claim.

2. **Comparisons to other zoom-then-behavior baselines.**

Zoomeye/Chain-of-Spot and guided-search methods aim at similar zoom-then-reason behavior. While the appendix notes a comparison in cost and accuracy,  the main paper doesn’t present head-to-head accuracy/latency plots vs. these strategies under identical token budgets. A direct comparison table would help isolate the value of feature replay vs. re-encoding crops [1-3]

3.  **Latency comparison.**

The paper provides a comparison on vision token cost, but as the method usually requires additional bounding box tokens and recall of the visual feature map, adding a inference time latency cost comparison compared with LLaVA-Next and other observe-then-reason baseline will help to understand the efficiency of the method.


[1]. Chain-of-Focus: Adaptive Visual Search and Zooming for Multimodal Reasoning via RL

[2]. Chain-of-Spot: Interactive Reasoning Improves Large Vision-Language Models

[3]. ZoomEye: Enhancing Multimodal LLMs with Human-Like Zooming Capabilities through Tree-Based Image Exploration.

**Questions:**

1. What fraction of generations emit at least one valid replay signal? What’s the average number of replay events per instance?

2. How often are boxes invalid or out of bounds?

---

> ### Author Response · Authors · 2025-11-25
> **Response to Reviewer ysjS**
>
> We are grateful to the Reviewer for your extensive review. We will address your questions point by point below:
>
> > W1 & Q2 — Replay supervision & parser brittleness.
>
> Thank you for raising the questions regarding replay supervision and parser brittleness. We address them in three parts.
> First, we measured the cases where the model failed to produce coordinate text between the special tokens that could be parsed by our rules, or produced invalid coordinates. Across various downstream benchmarks, the proportion of valid bounding boxes consistently ranges from 97.5% to 98.9%, indicating that only a few corner cases fail after VGR training.
> Second, we trained a variant of VGR-7B that uses absolute pixel coordinates instead of normalized (0–1) relative coordinates, and compared it with the original setting:
>
>  method| Box Type                      | DocVQA | MMStar | ChartQA | TextVQA | AI2D | RWQA
> ------------------|-------|--------------|--------|---------|---------|------|------
> VGR-7B | w/normalized    |      73.7   |  41.7   |  67.7  | 63.9 |    73.7|59.8
> VGR-7B  | wo/normalized  |     71.9       |  40.9   |  64.2  |  62.8  | 70.5 |  56.2
>
> Benefit from the global template construction enabled by LLaVA’s any-resolution design and the visual feature memory mechanism, the absolute-coordinate version performs only slightly worse than the original setting.
>
> Finally, using lmms-eval, we evaluate LLaVA-Next and VGR on RefCOCO, RefCOCOg, and RefCOCO+, reporting both the standard IoU>0.5 accuracy and the IoU metric. Results are shown below:
>
> Method                 | refcoco_acc@0.5 | refcoco_iou| refcocog_acc@0.5 | refcocog_iou| refcoco+_acc@0.5 |refcoco+_iou
> ----------------|----------------|------------------- |----------------|------------------- |--|--
> LLaVA-NeXT-7B            |     0.85  | 0.72 | 0.82        | 0.69 | 0.77|0.65
> VGR-7B            |  0.88    | 0.74  | 0.84      | 0.71| 0.78|0.68
>
> The results show consistent improvements over the baseline, suggesting that VGR training and data also enhance grounding performance.
>
> > W2 & W3 & Q1 — Comparison to zoom-then-reason baselines and valid replay signal.
>
> Thank you for your suggestions regarding the comparison between VGR and other zoom-then-reason approaches.
> First, Chain-of-Spot is built on LLaVA-1.5 rather than LLaVA-Next, and it resizes all images to a resolution of 334, whereas LLaVA-Next supports tiled processing of high-resolution inputs. This results in different LLM sequence lengths, making a direct comparison of inference latency unfair. Moreover, Chain-of-Spot is not training-free; its annotations were created only on LLaVA-1.5, preventing us from reproducing it on LLaVA-Next.
>
> For Chain-of-Focus, the inference cost has a different structure. It uses RL training, and the official evaluation requires an additional Qwen2.5-VL-72B model as a judge. Furthermore, due to the slow rollout stage, vLLM is used for inference acceleration, while other baselines are not adapted to vLLM. The official repository only provides evaluation scripts for the V* benchmark, so we conducted the comparison here:
>
> Backbone |Method                 | V* Bench | Inference Time|
> ------------|--------------|----------------|-------------------
> LLaVA-NeXT-7B     |   LLaVA-NeXT    |     56.4  | 1.04s |
> LLaVA-NeXT-7B | Zoom Eye     |  71.7  |  48.5s  |
> LLaVA-NeXT-7B|VGR     |  67.7    | 7.5  s |
> Qwen-2.5-VL-7B|CoF     |  88.0    | 3.6  s |
>
> Chain-of-Focus uses images around 2000×2000 resolution, and its inference framework reports a generation speed of ~80 tokens/s. We measured VGR’s generation speed as ~15 tokens/s, based on H20 GPUs.
> We also computed the average number of generated tokens and replay counts for VGR-7B across benchmarks:
>
>  Metric |MMStar               | V* Bench | ChartQA| DocVQA
> --------------------------|--------------------------|----------------|-------------------|--
> Repaly Times     |  2.0   |     1.7 |  2.4 |  2.1|
> Generate Tokens  |  98     |  110  |  155  |  113|
>
> Since VGR performs visual feature replay and produces longer outputs, it naturally incurs more inference time. Nonetheless, with the rapid development of frameworks such as vLLM and sglang, more efficient LLM inference is becoming increasingly feasible, and we plan to explore these directions.
>
> Finally, as shown in Table 7 (rows 3 and 4), the crop-based and memory-based approaches achieve comparable downstream performance, with the memory-based approach showing slight advantages. Crop-based methods also introduce additional ViT computation and increase the number of visual tokens passed to the LLM. Given the similar performance, we therefore choose the more computationally efficient memory-based approach, which better aligns with LLaVA-Next’s original dynamic-resolution design.
>
> If you have any further questions or concerns, please feel free to contact us at any time. We are always available and looking forward to further discussions with you. :)
>
> Best regards,
>
> All Authors

---

### Official Review · Reviewer_R29m · 2025-10-31

**Soundness:** 3
**Presentation:** 3
**Contribution:** 3
**Rating:** 4
**Confidence:** 5

**Summary:**

This paper proposes VGR (Visual Grounded Reasoning), a novel multimodal large language model (MLLM) designed to address language bias and limited visual detail understanding in existing multimodal chain-of-thought (CoT) reasoning approaches. VGR introduces a dynamic visual memory replay mechanism that enables the model to actively detect and retrieve visual tokens from key image regions during reasoning, mimicking human visual cognition. The authors also construct a large-scale supervised fine-tuning dataset (VGR-SFT) through a three-stage pipeline (cold-start generation, reject sampling, and annotation model scaling) to teach the model visual grounding and reasoning integration. Extensive experiments on benchmarks like MMStar, ChartQA, and AI2D show that VGR outperforms the LLaVA-NeXT-7B baseline with only 30% of the image tokens, validating its efficiency and effectiveness in visual-linguistic reasoning.

**Strengths:**

(1) The proposed dynamic visual memory replay mechanism is innovative, as it enables on-demand retrieval of key visual regions during reasoning, effectively mitigating language bias and enhancing fine-grained visual understanding.

(2) The VGR-SFT dataset construction pipeline is rigorous and scalable, avoiding manual annotation bias through model-generated grounding areas and ensuring data quality via reject sampling and rewriting.

(3) The experimental design is comprehensive, including extensive comparisons with state-of-the-art models, ablation studies on core components (e.g., detection loss, visual memory replay), and efficiency analysis, providing solid evidence for the method's superiority.

**Weaknesses:**

(1) The choice of the 72B MLLM for cold-start data generation lacks comparative analysis; the authors do not explain why smaller models (e.g., 13B or 34B) were not considered, nor do they demonstrate the advantages of the 72B model in generating initial visual grounding data.

(2) The use of Doubao1.5-VL's API for reject sampling lacks detailed parameter settings (e.g., temperature, top-p, response timeout) and verification criteria, making it difficult for readers to reproduce the data filtering process.

(3) The setting of β=2 in the detection loss (combining L1 and GIoU loss) is not supported by ablation experiments; the authors fail to test other β values (e.g., 1, 3, 4) to verify whether 2 is the optimal choice.

(4) The comparison with ZoomEye is incomplete, as it only reports results on V* Bench and HR-Bench 8K, lacking performance data on other key benchmarks (e.g., TextVQA, InfoQA) and detailed analysis of reasoning logic differences between the two methods.

(5) The combination experiments of visual encoders and LLMs are limited; the authors only test a few combinations (e.g., Qwen2.5+SigLIP, Vicuna+CLIP) and do not explore more advanced encoders (e.g., InternViT-6B, EVA-CLIP) or LLMs (e.g., LLaMA 3), limiting the demonstration of the framework's generalizability.

(6) The rationale for setting the maximum number of cropped images to 64 during test-time token scaling is unclear; the authors do not explain whether this threshold is determined by empirical testing or theoretical analysis, nor do they show performance changes when exceeding this threshold.

(7) The contribution of each subset in the VGR-SFT dataset (e.g., AI2D, GQA, ChartQA) to the model's overall performance is not analyzed; it is impossible to determine which data types drive the performance improvement, hindering the understanding of dataset design rationality.

(8) The trigger mechanism for replay signal generation during inference is not detailed; the authors do not clarify how the model decides when to retrieve visual memory (e.g., based on semantic cues in the reasoning chain or statistical thresholds), leading to ambiguity about the core logic of dynamic replay.

(9) Key training hyperparameters are missing, such as batch size, number of training epochs, weight decay, and learning rate scheduling strategy for both pre-training and fine-tuning stages, which is critical for reproducibility.

(10) The model's performance on low-resolution images (e.g., below 336×336) is not evaluated; given that real-world images often have varying resolutions, this omission limits the assessment of the model's practical applicability.

(11) There is no comparison with cutting-edge MLLMs like GPT-4V, Gemini Pro Vision, or Claude 3 Opus; the authors only compare with open-source models, failing to demonstrate VGR's competitiveness against commercial state-of-the-art systems.

(12) The quantification of language bias reduction is insufficient; the authors claim to mitigate language bias but do not use specific metrics (e.g., bias score, fairness indicators) to measure the degree of reduction, making the claim lack objective support.

(13) The details of high-resolution cropping in visual memory pool construction are vague; the authors do not specify key parameters such as crop overlapping ratio, number of crops per image, and selection criteria for crop positions, affecting the reproducibility of the visual memory module.

(14) The annotation error rate of the VGR-SFT dataset is not reported; the authors do not explain how they identified and handled annotation errors during data curation, raising concerns about data quality.

(15) Inference speed on different hardware platforms (e.g., A100, RTX 3090, RTX 4090) is not provided; the authors only report average reasoning time per question on specific benchmarks, failing to reflect the model's deployment feasibility on resource-constrained devices.

(16) Performance in multi-turn reasoning scenarios is not tested; the authors only evaluate single-turn question answering, while real-world applications often require continuous reasoning based on previous interactions.

(17) The specific thresholds for format verification and correctness verification in reject sampling are unclear; for example, the ANLS threshold for closed-ended tasks and the semantic alignment score threshold for open-ended tasks are not specified, making it difficult to replicate the data filtering process.

(18) The detailed structure of the MLP in the detection head is missing; the authors do not mention the number of layers, hidden size, activation function, or output dimension, which is essential for understanding the region detection mechanism.

(19) Generalization ability on cross-domain images (e.g., medical images, remote sensing images, satellite images) is not evaluated; the model is only tested on conventional VQA and OCR datasets, limiting the assessment of its applicability to specialized fields.

(20) The basis for selecting the MLLM used in data rewriting is not explained; the authors do not compare different models (e.g., LLaMA 2, Mistral) for rewriting effectiveness, leading to uncertainty about the optimal choice for this task.

(21) The selection of 2×2 and 4×4 pooling strategies for visual token compression lacks theoretical support; the authors do not explain why these pooling sizes are chosen over others (e.g., 3×3), nor do they provide ablation results on pooling strategies.

(22) The impact of model parameter size (7B vs 13B) on performance is not analyzed in depth; the authors only report basic results but fail to discuss how parameter scaling affects the trade-off between performance and computational cost.

(23) The balance of the VGR-SFT dataset across different task types is not addressed; the authors do not clarify whether the data distribution is balanced (e.g., proportion of OCR vs. general VQA tasks) or how imbalance is handled if present.

(24) Metrics for evaluating the accuracy of region selection in dynamic visual memory replay are missing; the authors do not specify how they measure whether the model selects the correct key regions, making it difficult to assess the effectiveness of the grounding mechanism.

(25) The handling of duplicate samples in the training data is not explained; the authors do not mention whether duplicate samples exist, how they were detected, or how they were processed (e.g., removal, merging), which may affect training stability.

(26) Performance in few-shot learning scenarios is not tested; the authors only use full-scale training data, failing to demonstrate the model's ability to adapt to low-data regimes, which is important for real-world applications.

(27) The comparison with Chain-of-Spot is insufficient; the authors do not discuss differences in reasoning logic, computational complexity, or performance on different task types, limiting the understanding of VGR's advantages over similar interactive reasoning methods.

(28) Latency introduced by visual memory replay is not discussed; the authors do not quantify the additional time cost of retrieving and processing visual tokens, which is critical for real-time applications.

(29) Interpretability analysis is limited; beyond region annotation, the authors do not provide other interpretability methods (e.g., attention visualization, reasoning chain decomposition) to explain how the model integrates visual and linguistic information.

(30) Detailed license information for datasets is missing; the authors only state that datasets are publicly available but do not specify license types (e.g., MIT, CC BY-SA) or any restrictions on use, raising potential copyright concerns.

(31) The fusion method of pre-training data (LLaVA-558K) and fine-tuning data (LLaVA-NeXT-770K + VGR-SFT) is not clarified; the authors do not explain whether the data is concatenated, mixed in batches, or processed with different weights, affecting reproducibility.

(32) Robustness on complex scenarios (e.g., occluded images, blurred images, low-light images) is not evaluated; the authors only test on standard datasets, failing to demonstrate the model's ability to handle real-world noise.

(33) The normalization method of bounding box coordinates in GIoU loss is not explained; the authors do not specify whether coordinates are normalized to [0,1] based on image size or other standards, leading to ambiguity in loss calculation.

(34) The relevance between visual cues and language reasoning in the dataset is not evaluated; the authors do not measure how well the annotated visual regions align with the reasoning chain, affecting the assessment of dataset quality.

(35) Training time and computational resource consumption are not reported; the authors do not specify GPU hours, number of GPUs used, or total training time, making it difficult for researchers with limited resources to replicate the work.

(36) The potential of combining VGR with RL methods (e.g., GRPO) is not explored; the authors only use supervised fine-tuning, failing to discuss whether RL can further enhance the model's reasoning and grounding capabilities.

(37) The impact of reasoning chain length on performance is not analyzed; the authors do not test whether longer or shorter reasoning chains affect accuracy or efficiency, limiting the understanding of optimal reasoning chain design.

(38) The generalization ability of the annotation model during dataset scaling is not evaluated; the authors do not measure how well the 14B annotation model performs on unseen data types, raising concerns about dataset quality during scaling.

(39) The impact of reducing visual tokens by 70% on the model's ability to capture complex visual information is not discussed; the authors do not clarify whether the token reduction leads to information loss in complex scenes (e.g., dense objects, complex layouts).

(40) Performance in multilingual scenarios is not tested; the authors only use English questions and images, failing to demonstrate the model's applicability to non-English languages, which is important for global use.

**Questions:**

*To facilitate discussions during the Rebuttal phase, authors are advised to respond point-by-point (indicating the question number).*

(1) Could you provide a comparative analysis of different cold-start models (e.g., 13B, 34B, 72B MLLMs) to justify why the 72B model was chosen for initial data generation? Please include metrics such as data quality (e.g., grounding accuracy), generation speed, and reject rate.

(2) What are the specific parameter settings (e.g., temperature, top-p, max response length) and verification criteria of Doubao1.5-VL used in the reject sampling pipeline? Could you provide the exact prompts used for format verification, correctness verification, and visual grounding verification?

(3) Why is β set to 2 in the detection loss? Could you conduct ablation experiments with different β values (e.g., 1, 3, 4) and present the results to confirm that 2 is the optimal choice?

(4) Could you extend the comparison with ZoomEye to more benchmarks (e.g., TextVQA, InfoQA, DocVQA) and provide detailed metrics including accuracy, reasoning time, and visual token usage? Also, please analyze the fundamental differences in reasoning mechanisms between VGR and ZoomEye.

(5) Have you tested more advanced visual encoders (e.g., InternViT-6B, EVA-CLIP-18B) or LLMs (e.g., LLaMA 3 8B/70B, Mistral 8X7B) with the VGR framework? If so, please provide the experimental results; if not, please explain the reasons and discuss the potential impact on performance.

(6) What is the rationale for setting the maximum number of cropped images to 64 during test-time token scaling? Could you present performance curves with varying numbers of crops (e.g., 20, 40, 64, 80) to demonstrate the trade-off between token count and performance?

(7) Could you analyze the contribution of each subset in the VGR-SFT dataset (e.g., AI2D, GQA, ChartQA) to the model's performance on different benchmarks? Please present ablation results showing the model's performance when trained on individual subsets.

(8) How does the model determine when to generate the replay signal during inference? Please explain the trigger mechanism in detail, including whether it relies on semantic cues, statistical thresholds, or other factors, and provide illustrative examples.

(9) Could you provide complete training hyperparameters, including batch size, number of training epochs, weight decay, learning rate scheduling (e.g., warm-up steps, decay rate), and optimizer type (e.g., AdamW, SGD) for both pre-training and fine-tuning stages?

(10) Have you evaluated the model's performance on low-resolution images (e.g., 224×224, 112×112)? Please present the results and discuss how resolution affects the model's grounding and reasoning capabilities.

(11) Could you provide a comparison with commercial state-of-the-art MLLMs (e.g., GPT-4V, Gemini Pro Vision, Claude 3 Opus) on the same benchmarks? If direct comparison is not feasible, please explain the constraints and provide indirect evidence (e.g., relative performance to open-source baselines vs. commercial models).

(12) How do you quantify the reduction of language bias in VGR? Please define specific metrics (e.g., bias score, fairness gap) and present comparative results between VGR and the LLaVA-NeXT baseline on these metrics.

(13) Please detail the high-resolution cropping strategy in visual memory pool construction, including parameters such as crop size, overlapping ratio, number of crops per image, and selection criteria for crop positions. Could you also explain how this strategy balances detail preservation and computational efficiency?

(14) What is the annotation error rate of the VGR-SFT dataset? How did you detect and handle these errors during data curation? Please provide statistics on error types (e.g., incorrect bounding boxes, misaligned labels) and their distribution across datasets.

(15) Could you report the model's inference speed (e.g., tokens per second, questions per minute) on different hardware platforms (e.g., A100, RTX 3090, RTX 4090, CPU)? Please also include memory consumption (e.g., GPU VRAM usage) for different model sizes (7B, 13B).

(16) Have you tested the model's performance in multi-turn reasoning scenarios? Please design a set of multi-turn tasks (e.g., sequential visual reasoning, follow-up questions) and present the results, including accuracy and consistency across turns.

(17) What are the specific thresholds for format verification and correctness verification in reject sampling? For example, what ANLS threshold is used for closed-ended tasks, and what semantic alignment score threshold is used for open-ended tasks? Please justify these threshold choices.

(18) Please provide the detailed structure of the MLP in the detection head, including the number of layers, hidden size, activation function, input/output dimensions, and initialization method. Could you also explain how this structure is optimized for bounding box regression?

(19) Have you evaluated the model's generalization ability on cross-domain images (e.g., medical images from ChestX-ray14, remote sensing images from NWPU-RESISC45)? Please present the results and discuss the challenges of adapting VGR to specialized domains.

(20) What is the basis for selecting the MLLM used in data rewriting? Could you compare the rewriting effectiveness of different models (e.g., LLaMA 2 7B, Mistral 7B, Qwen2 7B) using metrics such as reasoning coherence, format compliance, and ground-truth alignment?

(21) Why are 2×2 and 4×4 pooling strategies chosen for visual token compression? Could you conduct ablation experiments with other pooling sizes (e.g., 3×3, 5×5) and present results on performance, computational cost, and information preservation?

(22) How does the model's parameter size (7B vs. 13B) affect the trade-off between performance and computational cost? Please provide detailed metrics including training time, inference speed, memory usage, and accuracy across benchmarks for both sizes.

(23) Is the VGR-SFT dataset balanced across different task types? Please provide a detailed breakdown of data distribution (e.g., percentage of OCR, general VQA, science QA tasks) and, if imbalanced, explain how you addressed it (e.g., oversampling, weighted loss).

(24) What metrics do you use to evaluate the accuracy of region selection in dynamic visual memory replay? Please define the metrics (e.g., IoU with ground-truth regions, precision/recall of key region selection) and present comparative results between VGR and the baseline.

(25) How did you handle duplicate samples in the training data? Please explain the detection method (e.g., hash-based, semantic similarity) and processing strategy (e.g., removal, merging), and provide statistics on the number of duplicates found and processed.

(26) Have you tested the model's performance in few-shot learning scenarios? Please design experiments with varying training data sizes (e.g., 1K, 10K, 50K samples) and present results on key benchmarks, comparing VGR with other few-shot multimodal models.

(27) Could you provide a more in-depth comparison with Chain-of-Spot, including reasoning logic, computational complexity (e.g., number of forward passes), performance on spot-based visual search tasks, and adaptability to high-resolution images?

(28) Does visual memory replay introduce additional latency during inference? If so, could you quantify the latency overhead (e.g., percentage increase in reasoning time) and discuss potential optimizations (e.g., precomputing visual tokens, parallel processing)?

(29) Besides region annotation, could you provide other interpretability analyses (e.g., attention heatmaps, reasoning chain decomposition, visual-linguistic alignment scores) to explain how the model integrates visual and linguistic information for reasoning?

(30) Could you provide detailed license information for all datasets used in the study (e.g., LLaVA-558K, VGR-SFT subsets)? Please specify any restrictions on use, redistribution, or commercial application to address copyright concerns.

(31) How are the pre-training data (LLaVA-558K) and fine-tuning data (LLaVA-NeXT-770K + VGR-SFT) fused during training? Please explain whether the data is concatenated, mixed in batches, or assigned different weights, and justify the chosen method.

(32) Have you evaluated the model's robustness on complex scenarios (e.g., occluded images, blurred images, low-light images)? Please use standard robustness benchmarks (e.g., ImageNet-C, VQA-C) and present results on accuracy degradation compared to clean images.

(33) Please explain the normalization method of bounding box coordinates in the GIoU loss calculation. Are coordinates normalized to [0,1] based on the original image size, cropped patch size, or other standards? Please provide a mathematical formulation if applicable.

(34) How do you evaluate the relevance between visual cues and language reasoning in the VGR-SFT dataset? Please define a relevance metric (e.g., semantic similarity between region labels and reasoning steps) and present statistics on the dataset's relevance distribution.

(35) Could you report the detailed computational resource consumption for training VGR, including the number of GPUs used (e.g., 8×A100), total GPU hours, power consumption, and training time for pre-training and fine-tuning stages separately?

(36) Have you explored combining VGR with reinforcement learning (RL) methods like GRPO? Please provide preliminary results (if any) on whether RL can further improve the model's reasoning accuracy, grounding precision, or efficiency. If not, please discuss the technical challenges and potential benefits.

(37) How does the length of the reasoning chain affect the model's performance? Could you conduct ablation experiments with varying reasoning chain lengths (e.g., short, medium, long) and present results on accuracy, efficiency, and language bias reduction?

(38) How do you evaluate the generalization ability of the 14B annotation model during dataset scaling? Please present metrics such as grounding accuracy, reasoning coherence, and reject rate on unseen data types compared to the cold-start 72B model.

(39) Does reducing visual tokens by 70% lead to information loss in complex visual scenes (e.g., dense objects, complex layouts)? Please design experiments with such scenes and present comparative results between VGR and the baseline (using full tokens) on detail-rich benchmarks.

(40) Have you tested the model's performance in multilingual scenarios? Please evaluate VGR on non-English benchmarks (e.g., VQA in Chinese, German OCR tasks) and present results on accuracy, grounding precision, and language bias compared to English-only performance.

---

> ### Comment · Reviewer_R29m · 2025-11-13
> **To Readers and Authors**
>
> I observed a discussion regarding my review comments on Xiaohongshu, hence I am issuing a response herein:
>
> 1. All these comments are the result of the reviewer's dedicated time and rigorous effort, composed meticulously word by word; they are not generated by AI tools such as GPT.
>
> 2. Both the authors and readers may have mistakenly assumed there are 80 comments. Please read the paper and the review with due diligence! There are distinctly 40. In response to 40 weaknesses, the reviewer proposed 40 corresponding questions. The reviewer also emphasized to the authors at the outset the necessity of providing point-by-point responses (with clear indication of the question serial numbers).
>
> Furthermore, unrelated to the content of this manuscript, I wish to share several reflections:
>
> 1. In the current impetuous and intricate society, if one aspires to be a scholar, it is imperative to attain inner calm. Scientific research demands tranquility, particularly peace of mind. Do not let yourself be unable to even clarify how many comments require a response.
>
> P.S. This constitutes the primary motivation for my writing this passage. I hope the authors will not be distracted by external disruptions and erroneously perceive there to be 80 comments.
>
> 2. What is the objective of writing and submitting a paper? Is it merely for acceptance and publication? Or is it to enhance one's academic proficiency, for instance, the quality of the manuscript?
>
> 3. If your initial response to receiving review comments is to disregard them or even be apathetic toward their content; if you feel psychologically unbalanced solely because the reviewer raised a substantial number of questions, then I pose the question: For those who hold such attitudes, what is the purpose of submitting papers? Is it for the advancement of academia?

---

> > ### Author Response · Authors · 2025-11-13
> >
> > Dear Reviewer R29m:
> >
> > 1. We are actively expediting the preparation of supplementary experiments and supporting materials to address the comments raised by the five reviewers.
> >
> > 2. We have observed that responses impersonating the authors of this paper have appeared in the public comment section. To preclude any potential misunderstandings, we hereby formally clarify: all authors of this paper have strictly adhered to the double-blind review protocols and did not publicly conduct any discussions related to this review process during the rebuttal period via any social media platforms.
> >
> > 3. Furthermore, we are not privy to the specific details of the alleged discussions on the 'Xiaohongshu' that you referenced.

---

> > > ### Comment · Reviewer_R29m · 2025-11-13
> > > **RE: Official Comment by Authors**
> > >
> > > Thank you for your reply and clarification!
> > >
> > > 1. Please continue with the preparation of your reply and experiments.
> > >
> > > 2. I have observed someone impersonating, hence I am issuing this response.
> > >
> > > 3. Pay no attention to those "distractions", devote yourself wholeheartedly to preparing your reply and experiments. I hope their interference will not hinder your progress! Keep up the good work!

---

> > ### Public Comment · ~Hengrui_Zhang5 · 2025-11-18
> >
> > In the current impetuous and intricate society, if one aspires to be a scholar, it is imperative to attain self conciousness. Do not let yourself be only able to write "by hand" every word that your AI master tells you to and mistake it as self crafted.

---

> > ### Public Comment · ~Liu_Jingxin1 · 2025-11-27
> > **Does your behavior boost the advancement of academia?**
> >
> > Will a mentally normal person can think out 80 weaknesses and questions for a paper?? Could you please show your name? I think you will be popular on the social media.

---

> ### Public Comment · ~Qijia_Shen1 · 2025-11-14
> **This reviewer's question is completely AI generated verified with gpt-zero**
>
> Just ran this reviewer's (R29m) questions through the gpt-zero, which turns out to be 100% AI generated. Everyone can replicate it on gpt-zero themselves.
>
> This has constituted a serious academic misconduct, and should not be tolerated.

---

> ### Public Comment · ~Zeyuan_Feng1 · 2025-11-15
> **Why don't you write a paper based on your comments**
>
> This is the most shameless reviewer I have ever seen. I will be extremely disappointed if the AC even considers these comments for a millisecond.
>
> Almost every single one of the 40 so-called “weaknesses” is just asking for another ablation study or baseline comparison. This reviewer doesn't understand at all that a research paper only needs to clearly present and justify its main contribution, but instead asks for a product development report where every microscopic detail must be thoroughly studied.
>
> Yet, this reviewer tries to lecture other people about being professional, claiming that the core purpose of its peer review "is to help authors enhance the academic quality of their papers or research". If this reviewer still has its final little "academic integrity", then please proudly follow its own academic standard for every paper it write from now on. I'm very excited to witness its massive, comprehensive, encompassing, extensive, panoptic academic production in the future.

---

> ### Public Comment · ~Talia_Ringer1 · 2025-11-15
> **Two other papers with the same review pattern**
>
> 1. https://openreview.net/forum?id=GlXyFjUbfN
>
> 2. https://openreview.net/forum?id=8qk6eUnvbH
>
> I hope the AC will look into this. Always 40 weaknesses, 40 questions, mostly arbitrary and requiring new experiments. Always the same score (4) and same confidence (5). Clearly not reasonable, at the very least.

---

> ### Public Comment · ~BelthOris1 · 2025-11-16
> **ICLR is truly blessed to have found such a discerning reviewer**
>
> The review platform has become a personal stage for your monologue, delivered through your "first and final public response." While researchers the world over may be pursuing "peace of mind," you alone seem to find transcendent spiritual fulfillment in profound inquiries like, "AHHHHH, why was my mom's birthday was not used for the $\beta$ parameter in the detection loss?"
>
> While everyone else may be engrossed in creating work that “demands tranquility, especially peace of mind,” you stand apart. By all means, we encourage you to share this unique philosophy far and wide:
> Even if an ICLR paper experimentally demonstrated results that surpass GPT-4V, you are free to cry foul and accuse the authors of failing to steal the source code and run it on the same hardware as open-source models.
> Even if they showed their method works on an RTX 3060Ti laptop, you can still demand they run it on a Raspberry Pi and an IBM quantum computer—simultaneously.
> You have no regard for others' peace of mind. Your goal is not understanding, but the hollow victory of proving others wrong.
> THIS is the true, hidden path to scholarly enlightenment.

---

> ### Public Comment · ~Matúš_Pikuliak1 · 2025-11-16
>
> This review is a clear example of academic misconduct. We cannot tolerate overwhelming authors with AI slop. I think it's reasonable to expect that this reviewer will be banned from the conference.

---

> ### Public Comment · ~Andreas_Kirsch1 · 2025-11-16
> **AI-generated questions**
>
> I've verified this with Pangram's checker (and other checkers).
>
> They've tested all ICLR 2022 reviews using their AI checker and report a 0% FPR for fully AI generated reviews:
>
> https://x.com/max_spero_/status/1989923081993809973
>
> The questions here seem to be fully AI generated:
>
> https://www.pangram.com/history/911cec8a-7f6f-42e2-b0cb-f4bc7eac6e5b/?ucc=f3LwxfWdWJI
>
> ---
>
> See also: https://iclr.pangram.com/reviews?submission_number=23089

---

> ### Public Comment · ~Adrian_Kucia1 · 2025-11-16
> **AI-Generated Reviews**
>
> Based on the statistics from (https://iclr.pangram.com/submissions?query=&submission_number=&sort_by=submission_id_hash&sort_dir=asc&page=3&ai_content_filter=&avg_rating_filter=)
>
> 1% of fully generated reviews have a avg. rating of 2.90 . There is 1% (199) of reviews which are 90%-100% AI generated.
>
> However, the reviews which are generated by human 0-10% AI have avg. rating of 4.36.
>
> Many of those reviews make no sense..."Lack of ablation study" or "lack of benchmark". This all has been included in main body of the paper.
>
> It states that : "reviewers who submit low quality reviews and fail to improve them upon being warned by ACs may have their own papers desk rejected: Low quality reviews (e.g., placeholder reviews) will be flagged by ACs and SACs, and the flagged reviewers will be warned and urged to update the review. Reviewers who do not respond to these warnings will be liable to having their own papers desk rejected."

---

> ### Public Comment · ~Michael_Lanier1 · 2025-11-16
> **AC should disregard this review**
>
> I encourage the AC to disregard this review during final reviews. While some of this is reasonable (eg where is the beta ablation), a request for a hardware ablation indicates that this review was either AI generated or the reviewer is not well read enough in this field (or perhaps insufficiency skilled) to be making the review in the first place.

---

> ### Comment · Reviewer_R29m · 2025-11-17
> **To Readers and Authors (2)**
>
> Dear Readers and Authors,
>
> **I would like to reaffirm that this review is crafted entirely by hand, with every word written by myself, not generated by AI.**
>
> AI detection tools are incapable of fully distinguishing between human-authored and AI-generated content.
>
> As specified in the ICLR 2026 Reviewer Guide (https://iclr.cc/Conferences/2026/ReviewerGuide), initial reviews are solely intended to help authors improve the quality of their papers and will not affect the final decision score.
>
> Considering the authors are diligently working on their rebuttals and responses, and may be impacted by these public comments, this will be my final statement on the matter. I will not respond to any further inquiries from readers going forward.
>
> Thank you all for your attention!
>
> R29m

---

> ### Public Comment · ~Jiache_lu1 · 2025-11-17
> **Appeal for Desk Rejection of All Papers Submitted by Reviewer R29m to ICLR 2026**
>
> **I appeal that all the papers Reviewer R29m submitted to ICLR 2026 should be desk rejected**, based on the following critical concerns regarding their unprofessional review practices:
>
> First, Reviewer R29m claims their reviews are entirely handcrafted, yet their evaluations exhibit highly suspicious uniformity: **all reviews consistently include 40 weaknesses, 40 questions** (most of which are arbitrary), and rigidly demand new experiments. Additionally, **every review assigns the exact same score (4)**.
> Two representative examples are linked below:
> 1. https://openreview.net/forum?id=GlXyFjUbfN
> 2. https://openreview.net/forum?id=8qk6eUnvbH
>
> Second, **talk is cheap, show us the proof**. Reviewer R29m must provide concrete proof that their reviews are not generated by large language models (LLMs). If nearly all LLM detection tools indicate the reviews are LLM-generated, their assertion of innocence lacks credibility.
>
> Given the above unprofessional and questionable review behaviors, I urge Area Chairs (ACs) and Senior Area Chairs (SACs) to thoroughly investigate Reviewer R29m’s reviews. **I reiterate my appeal for the desk rejection of all papers submitted by Reviewer R29m to ICLR 2026**.

---

> ### Public Comment · ~Wenqi_Guo3 · 2025-11-17
> **If it's not AI generated, then it means it's low quality**
>
> The review contains several signs that appear inconsistent with the standard world model. For example, requesting benchmarks across multiple hardware platforms does not reflect common experimental practice. It doesn't matter if the review is generated by AI or written by R29m (and since this is hard to proof), it has a mismatch raises concerns about the practical validity of the review comments.

---

> ### Public Comment · ~Size_Wu1 · 2025-11-18
> **A Big Thanks to Reviewer R29m**
>
> Everyone except the authors of Submission23089 owes a big thanks to Reviewer R29m for the absolute comedy show they put on. Seriously, crafting 80 points of bullshits by hand can not be easy. Honestly, I couldn’t stop laughing while reading through questions (15), (18), (32), and the rest of the absurd list. If this review wasn’t a joke, I don’t know what is.
>
> Unlike Reviewer R29m, who clearly has a knack for spewing whatever nonsense comes to their mind, without help from an LLM, even though the detection rate is 100%, I’ll admit this comment was polished, shall we say 'refined,' by GPT. My command of the English language just can’t do justice to the kind of ridiculousness in those points, so I needed a little help from AI to express how utterly nonsensical it all was.
>
> Honestly, though, maybe we should thank R29m for turning an otherwise soul-crushing review cycle into something we can laugh about. Who knew that 80 points of random bullshit could be the highlight of our week? It’s almost like he was trying to lighten the mood in an academic environment that takes itself way too seriously. So yeah, big shoutout to R29m for the laughs… everyone else, feel free to join in, just don’t take the review seriously.
>
> R29m could have just left nothing but a zero rating, but he was so nice and brought as many as 80 jokes!

---

> ### Public Comment · ~Shuai_Zhao1 · 2025-11-19
> **Nitpicking Reviews**
>
> ```
> (2) The use of Doubao1.5-VL's API for reject sampling lacks detailed parameter settings (e.g., temperature, top-p, response timeout) and verification criteria, making it difficult for readers to reproduce the data filtering process.
>
> (15) Inference speed on different hardware platforms (e.g., A100, RTX 3090, RTX 4090) is not provided; the authors only report average reasoning time per question on specific benchmarks, failing to reflect the model's deployment feasibility on resource-constrained devices.
> ```
>
> It appears that most reviews are just nitpicking. Whether it is AI-generated or human-written, receiving such reviews is a nightmare.

---

> ### Author Response · Authors · 2025-11-25
> **Response to Reviewer R29m(1/3)**
>
> We are grateful to the Reviewer for your extensive review. According to AC's suggestions, we will respond to your first 10 issues raised point by point below:
> >  W1&Q1: Different cold-start model size.
>
> Because VGR-SFT requires the model to generate both high-quality grounding coordinates and long-CoT reasoning process, the cold-start model must possess sufficiently strong grounding ability and foundmetal ability. Qwen-VL-2.5 has both of these capabilities and is open source. We evaluated the Qwen2.5-VL family and found that the smaller variants (7B/30B), even with n-shot prompting, failed to follow our detailed instructions in roughly 70% of cases, producing incorrect long-CoT structures or grounding outputs. In contrast, the 72B model followed our instructions reliably in most cases, which is the reason we selected it for the initial data generation. Since serving the 72B model brings high inference costs, we subsequently trained a smaller 14B annotation model using the first batch of data to improve the overall pipeline efficiency. These details are described in Section 4.3 and Appendix D.1.
>
> >  W2&Q2: Doubao1.5-VL parameter settings and prompts.
>
> We did not tune the parameters of Doubao1.5-VL and instead used the official default settings, with a maximum response length of 8192. The exact prompts used for format verification, correctness verification, and grounding verification are fully provided in Appendix D.1.
>
> >  W3&Q3: β values in detection loss.
>
> The choice of β=2 follows the standard hyperparameter setting commonly used in detection tasks, as in DETR [1]. Therefore, we did not conduct additional ablation experiments on alternative β values.
>
> >  W4&Q4: Compare with ZoomEye.
>
> ZoomEye is a training-free method that abstracts the image into a tree structure and performs rule-based tree search. The bounding boxes used during inference are predetermined by the search process.
> In contrast, VGR enables native free-form visual reasoning. The model can autonomously trigger visual replay at any stage of reasoning and reuse visual memories in a human-like cognitive flow. Unlike multi-turn or manual-cropping approaches, VGR embeds grounding capabilities directly into the model.
>
> ZoomEye is specifically designed for high-resolution tasks, visually dense benchmarks, and its paper only evaluates VBench, HRBench, and MME-RealWorld. It does not evaluate TextVQA, InfoQA, or DocVQA, nor is it intended for these domains. Therefore, extending the comparison to such benchmarks would not be a fair evaluation.
> Our detailed comparison with ZoomEye on the relevant high-resolution benchmarks (VStar Bench and HRBench) is reported in Table 10 in Appendix A.2.
>
> >  W5&Q5: Larger ViT and larger LLM.
>
> Table 3 in our main paper already validates VGR's effectiveness across multiple visual encoders (e.g., SigLIPSO400M/14@384, InternViT-300M/14@448-v2.5) and LLMs (e.g., Qwen2.5-7B-Instruct). The larger encoders you mentioned (e.g., InternViT-6B, EVA-CLIP-18B) and larger LLMs (e.g., LLaMA 3 70B, Mistral 8×7B MoE) require substantially higher computational and infrastructure costs, and we have not yet explored adapting VGR training to MoE architectures.
>
> We believe that validating VGR on widely adopted baselines such as LLaVA-NeXT and further demonstrating its effectiveness across multiple LLMs and encoders is sufficient to support our contributions. Additional experiments with significantly larger models would not change our conclusions.
>
>
> [1] Carion, Nicolas, et al. "End-to-end object detection with transformers." European conference on computer vision. Cham: Springer International Publishing, 2020.

---

> ### Author Response · Authors · 2025-11-25
> **Response to Reviewer R29m(2/3)**
>
> >  W6&Q6: Test-time scaling.
>
> Thank you for the insightful suggestion on further analyzing the effect of different crop limits during test-time token scaling. In our design, the maximum number of cropped regions during training is fixed at 20, while the test-time limit is extended to 64. This extension is intended solely to examine whether allowing more visual tokens at inference can yield additional gains, particularly for extremely high-resolution images.
>
> However, we believe that performing a full performance curve over 20/40/64/80 crops would bring limited practical value, for several reasons:
> Most benchmark images do not produce a large number of crops. Only a small subset of ultra-high-resolution samples (e.g., in HRBench or VStar) reach crop counts near 64. Increasing the limit beyond 64 has a negligible impact on the overall benchmark results.
>
> Excessive visual tokens can push the sequence length beyond the native context window of Vicuna/LLaVA-NeXT, leading to performance degradation. This introduces a new confounder: the curve would reflect not only “crop count vs. performance,” but also “context overflow vs. degradation,” making the analysis less meaningful.
> The model is trained with a maximum of 20 crops, and extending to 64 already covers a substantial train–test resolution mismatch. Further extending to 80 or more would place the model far outside its training distribution, making results unstable and difficult to interpret.
>
> We will include clarifications in the appendix and provide representative high-resolution examples to illustrate the behavior under different crop limits.
>
> >  W7&Q7: Ablating each subset in the VGR-SFT dataset.
>
> Thank you for the suggestion. We fully understand the motivation behind evaluating the contribution of each dataset subset (e.g., AI2D, GQA, ChartQA). However, conducting such per-subset ablations is infeasible or would not yield meaningful conclusions in our setting, for the following reasons:
>
> To our knowledge, existing multimodal CoT and reasoning works (e.g., Visual CoT, ZoomEye, Chain-of-Focus, CogCoM) do not perform dataset-subset ablations in their SFT stage. This is largely due to the same challenge we face: the tasks and data sources are highly interdependent.
>
> The subsets in VGR-SFT are strongly complementary rather than separable.
> GQA and COCO mainly contain natural images; ChartQA, DVQA, and DocVQA contain chart and document data; AI2D incorporates visual reasoning for understanding scientific charts. Training on any single subset cannot teach the model the full VGR-style long-form reasoning behavior. As a result, per-subset training would not meaningfully reflect the “contribution” of that subset, because the overall capability is merged.
>
> The VGR-SFT data follows a unique long-chain VGR-style CoT format, which is fundamentally different from standard QA. When trained on small, isolated subsets, the model cannot sufficiently learn the VGR-style reasoning format, making it unable to follow the prompts during evaluation. In other words, per-subset ablation would fail to produce valid outputs and thus cannot be evaluated reliably.
> For these reasons, we believe that per-subset ablation would not offer interpretable insights.
>
> >  W8&Q8: Replay signal during inference.
>
> Section 3 provides a detailed explanation of the replay control signal, together with Figures 1 and 3. Here we briefly restate the mechanism:
>
> During training, each replay region is encoded as grounding coordinates [x1, y1, x2, y2], and the model retrieves the corresponding visual tokens when this signal appears.
>
> During inference, the model must autonomously decide when to trigger replay, guided by the patterns learned from VGR-SFT. The model outputs the replay coordinates using special tokens \<sot\> and \<eot\>. Our processor extracts these coordinates, retrieves the visual features from the original image, inserts them back into the context, and resumes decoding. The end-of-sequence token is removed to ensure that the model views this as continuous reasoning rather than multi-turn dialogue.

---

> ### Author Response · Authors · 2025-11-25
> **Response to Reviewer R29m(3/3)**
>
> >  W9&Q9: Complete training hyperparameters.
>
> As stated in Section 5 of our main text, we follow the exact hyperparameter settings of LLaVA-NeXT, training all data for one epoch. The learning rate is 1e-5 for the pre-training stage and 2e-5 for fine-tuning (Vicuna-7B). The ViT learning rate is set to one-tenth of the base rate. We use a warmup ratio of 0.03, a cosine scheduler, and zero weight decay.
>
> >  W10&Q10: Performance on low-resolution images.
>
> We evaluated VGR-7B on OCRBench and obtained a score of 545, slightly improving over the 535 baseline. However, the overall gain on low-resolution images is limited for the LLaVA-NeXT family because it does not adopt an AnyRes ViT like Qwen-VL.
> Low-resolution images (e.g., 224×224) are upsampled to 334×334 for CLIP-ViT, meaning the model already sees a sufficiently clear input, and additional visual replay may not provide further benefit.
> VGR’s strength is primarily demonstrated on chart-intensive datasets (ChartQA, DocVQA) and high-resolution benchmarks (VStar, HRBench), where local details are critical, and the original image resolution is often insufficient.
>
> If you have any further questions or concerns, please feel free to contact us at any time. We are always available and looking forward to further discussions with you. :)
>
> Best regards,
>
> All Authors

---

> > ### Comment · Reviewer_R29m · 2025-11-27
> > **RE: Response to Reviewer R29m**
> >
> > I would like to thank the AC and the author for their feedback. My original intention was to raise as many questions as possible: if the author had addressed them to the best of their ability, the score would have been 10 points. However, taking into account time constraints, appropriate deductions would be made if the method has inherent limitations.
> > Overall, the author has addressed some of my concerns, and I have decided to raise the score to 6 points.

---

> > > ### Public Comment · ~Lei_Wu5 · 2025-11-27
> > >
> > > Is this decision truly based on author's response or the fact that your identity is disclosed?

---

> ### Public Comment · ~Liu_Jingxin1 · 2025-11-27
> **Came From Xiaohongshu and see 40+ amazing review comments**
>
> Hello, everyone. I came from xiaohongshu and found there are many discussions about the amazing 40+ review comments. Most people admit this review is hot. I don’t believe there will be a person can think out so many reviews util I find this paper. So amazing!  This is very good, but why there is 40 review comments, not 100+ review comments??  Maybe it's a philosophical question.

---

### Official Review · Reviewer_CQgV · 2025-11-01

**Soundness:** 4
**Presentation:** 4
**Contribution:** 3
**Rating:** 8
**Confidence:** 3

**Summary:**

The paper introduces the VGR framework, designed to support visual reasoning through visual memory replay during the thinking process. The main contribution of VGR lies in its visual memory replay module, which retrieves and replays visual information from memory. During inference, the model can request projections of image patches from memory, enabling it to revisit relevant visual regions and perform more focused reasoning. This approach allows the model to start with a smaller set of snapshot embeddings (around 30% of the original LLaVa) without sacrificing quality.

To train VGR, the authors first use an existing model to “cold-start” a dataset: given an image and a question from existing benchmarks (e.g., AI2D), the model generates a reasoning chain and answer. A rejection strategy is then applied to filter out incorrect samples using formal verification, correctness checks, and visual grounding validation. A smaller model is trained on the verified cold-start data and later used as an annotator to create a large-scale dataset. This larger dataset is then used with SFT methods to train the final VGR model.

The evaluation section is well structured, consisting of two main parts: (1) main results demonstrating the strong performance of the VGR framework across a range of datasets — both those included in the cold-start data and new, out-of-domain benchmarks such as MMStar and TableVQA — and (2) an extensive series of ablation studies exploring different components of VGR, including its reasoning process, memory module, and loss function.

**Strengths:**

Overall, the paper is clearly written and methodologically solid. The assumptions are well stated, and the authors do an excellent job of detailing their experimental setup and reasoning process. The evaluation section is one of the strongest aspects of the work, especially the ablation study, which provides valuable insight and helps position the paper as a meaningful contribution to the field.

**Weaknesses:**

There are a few points that could use more clarification.
First, the paper doesn’t clearly separate in-domain and out-of-domain benchmarks in Table 2. I tried to cross-check the datasets listed in Table 1 myself to identify which ones were new, but I would still like the authors to explicitly confirm whether the evaluations indeed include out-of-domain data. Without this distinction, it’s hard to fully assess the framework’s generalization ability.

Second, the main comparisons focus on relatively smaller VLMs. It would be interesting to see how VGR performs when integrated with larger models. If there are challenges or limitations in doing so, it would be helpful for the authors to discuss them. Otherwise, including such results would strengthen the paper’s claims about scalability.

Finally, as shown in Table 10, VGR’s main drawback appears to be its computational cost (e.g., 1s runtime for LLaVa-NeXT vs. 7s for VGR). Apart from a brief note in Appendix A.2, the paper doesn’t explore this issue in depth. A more thorough discussion of runtime and hardware costs would make the evaluation more balanced and practical.

**Questions:**

Q1) Why did the authors choose not to formulate the memory access as a tool call, rather than introducing special tokens? To me, that seems like a potentially more straightforward approach.

---

> ### Author Response · Authors · 2025-11-25
> **Response to Reviewer CQgV**
>
> We are grateful to the Reviewer for your extensive review. We will address your questions point by point below:
>
> > W1: Separate in-domain and out-of-domain benchmarks in Table 2.
>
>  Thank you for your question regarding the separation of datasets in Table 2.
>  We clarify the data separation in Table 2 as follows. All images in VGR-SFT are sourced strictly from the LLaVA-NeXT training stage (LLaVA-NeXt-770K) without introducing external images. While LLaVA-NeXt-770K contains training data corresponding to some downstream benchmarks (e.g., AI2D, DocVQA, and ChartQA), other datasets within it (such as LLaVA-COCO, GQA, DVQA, and OCRVQA) do not appear in our evaluation benchmarks. We will add a clearer explanation of these distinctions to the main paper, and we hope this resolves your concerns.
>
> >  W2: Integrating VGR with larger models.
>
>  Thank you for raising the question about integrating VGR with larger models. Due to resource constraints, our main experiments focus on the widely used 7B models, within which we conducted comprehensive ablation studies and analyses to validate the effectiveness of the VGR framework and the VGR-SFT data. However, we also demonstrate scalability: Appendix B reports results on LLaVA-13B, and Table 3 shows consistent improvements across various LLM backbones and ViT architectures. These results demonstrate that VGR consistently improves performance across different visual encoders, base LLMs, and benchmarks. Following the submission, we will initiate training for a LLaVA-NeXT 34B version of VGR (along with its baseline). Because this will take more time, we plan to include the results in Appendix B of the camera-ready version.
>
> >  W3: Inference cost with VGR.
>
> We acknowledge the concern regarding inference cost (also raised by Reviewer ysjS). The increased latency stems primarily from two factors. First, LLM inference time is largely determined by the next token generation mechanism, making the runtime closely tied to output length. Our baseline LLaVA-NeXT does not perform any reasoning and only generates short answers, allowing it to complete inference quickly. In contrast, models using CoT or thinking-style answers must generate longer reasoning chains, which naturally increases latency. The inference times reported in the paper were measured using H20 GPUs and the Lmms-eval framework.
>
> Second, we computed the average number of generated tokens, replay times, run time for VGR-7B on different tasks, as shown in the table below:
>
>  Metric|MMStar| V* Bench| ChartQA| DocVQA
> -|-|-|-|-
> Repaly Times  | 2.0 | 1.7  |  2.4 |  2.1 |
> Generated Tokens  | 98 | 110 | 155|113|
> Run Time | 5.84s | 7.51s | 11.57s | 6.51s
>
> As shown, VGR requires visual feature replay and longer outputs during answering, which contributes to the increased inference time. Nonetheless, thanks to rapid developments in the open-source ecosystem—such as vLLM and sglang, a more efficient LLM inference framework is becoming increasingly accessible. We will continue to follow and adopt these technologies to further improve the inference efficiency of our model.
>
> > Q1:Why did the authors choose not to formulate the memory access as a tool call, rather than introducing special tokens? To me, that seems like a potentially more straightforward approach.
>
> Thank you for your insightful suggestion regarding the formulation of memory access. We agree that the tool-call method is a straightforward solution. In fact, the approach adopted by VGR can be regarded as a specialized form of tool call—for instance, the memory access tool is activated via a special token, and the retrieval result is passed to the network. We believe the design of VGR does not conflict with tool calls and can even be combined with other tools in future research. We also highlight that we provide a complete training pipeline for our strategy, which does not rely on the inherent capabilities of large language models (LLMs), whereas the tool-using capacity of many models stems from black-box training.
>
> To be more specific, let us compare VGR’s memory retrieval with the crop strategy, a very common tool used in other studies. Compared with the crop strategy, VGR’s visual memory retrieval is more consistent and efficient. All visual tokens for replay are precomputed only once, and the multi-level design of visual resolution (via different pooling levels) allows the model to allocate computation to the most critical regions. This design not only enables the free-form visual reasoning we mentioned but also is more efficient: it saves approximately 70% of tokens while outperforming the baseline. We compared our memory strategy with the crop strategy in Table 7 and found that the performance of the two strategies is similar, while the memory access is more efficient.
>
> If you have any further questions or concerns, please feel free to contact us at any time. We are always available and looking forward to further discussions with you. :)
>
> Best regards,
>
> All Authors

---

### Official Review · Reviewer_joFM · 2025-11-03

**Soundness:** 3
**Presentation:** 3
**Contribution:** 2
**Rating:** 4
**Confidence:** 4

**Summary:**

This paper introduces VGR (Visual Grounded Reasoning), a framework that aims to enhance multimodal reasoning in vision-language models by incorporating a visual memory replay mechanism. The key idea is to allow the model to explicitly reference relevant visual regions during the reasoning process through a controlled replay signal (<sot>[x1,y1,x2,y2]<eot>). This design reduces the number of visual tokens, improves efficiency, and enhances interpretability. The authors conduct extensive experiments on multiple benchmarks (e.g., AI2D, ChartQA, MMStar), showing consistent improvements over LLaVA-NeXT baselines with fewer visual tokens. The paper also provides detailed ablation studies and efficiency analyses. While the empirical results are strong, the conceptual novelty is somewhat limited, and certain aspects of the method and presentation could be improved.

**Strengths:**

1. The paper includes a large number of experiments across diverse benchmarks with fair comparison to strong baselines, along with clear ablations and efficiency metrics.

2. The proposed replay mechanism and token-efficient design are well implemented and practically useful for multimodal reasoning systems.

**Weaknesses:**

1. The work is technically solid; however, the contribution in terms of novelty could be clarified further. The concept of visual CoT (chain-of-thought grounded in image regions) has been explored in prior studies (e.g., VisualCoT[1], SketchPad[2], Refocus[3]), which are currently only briefly discussed. Including a more detailed comparison could help position the paper within the existing literature. Among these, approaches most similar to “visual CoT” could be highlighted more explicitly, both conceptually and empirically.
2. The paper adopts LLaVA’s any-resolution encoding to handle arbitrary image sizes. It would be helpful to provide additional discussion on why this design choice is preferred over an alternative strategy such as cropping the image into regions, which may be more architecture-agnostic and computationally straightforward. Clarifying this could strengthen the reader’s understanding of the method’s design trade-offs.
3. The title emphasizes “Visual Grounded Reasoning.” It may be useful to clarify the scope of reasoning in this context. Prior studies (e.g., DeepSeek-R1[2]) suggest that reasoning ability in large language models often arises from reinforcement learning or dedicated reasoning-phase training, whereas supervised fine-tuning primarily encourages pattern learning or imitation. Explicitly situating the current work within this context could help manage expectations regarding the type of reasoning capability demonstrated.
4. The proposed replay mechanism assumes that accurately predicting relevant visual regions contributes to improved reasoning. Currently, the paper does not examine how often the replay regions are correctly predicted, nor how reasoning performance is affected when the replay predictions are inaccurate. A quantitative analysis, such as tracking replay prediction accuracy during training and correlating it with task performance, could provide stronger evidence for the causal role of replay. This would also help determine whether the model genuinely relies on visual replay or mainly benefits from additional supervision signals.
5. This paper only conducts experiments on the LLaVA-Next backbone. It would be better to conduct experiments on other backbones, such as Qwen-VL or InternVL.

**Questions:**

Please refer to the Weaknesses.

---

> ### Author Response · Authors · 2025-11-25
> **Response to Reviewer joFM(1/3)**
>
> We are grateful to the Reviewer for your extensive review. We will address your questions point by point below:
> >   W1: Comparisons with related works.
>
> Thank you for your insights regarding the novelty of VGR. We are happy to discuss the comparison with these works and add it to the latter revision of our paper. We address three distinct novelties of VGR, which differ from previous attempts:
>
> - **Native Free-form Visual Reasoning**
>
>    VGR enables the VLM to retrieve and reuse visual memory dynamically. Compared with Visual CoT, which trains the model with a multiturn dialog with predefined forms, VGR enables the model to reuse visual memory during complex visual thinking. This allows VGR for flexible "visual re-reading" at any step of the reasoning chain, interleaving summary, reflection, and visual replay—without manual cropping interventions. In contrast, Visual CoT can only retrieve the cropped area once and can only think with a static path.
>
>    VGR eliminates the need for a multi-turn dialogue format to guide the model to first output bounding boxes, followed by manual cropping and re-input of relevant visual features and text. Instead, VGR's visual feature replay not only offers greater flexibility in format but also, more importantly, embeds grounding reasoning capabilities within the model itself, enabling the model to spontaneously inspect highly relevant image regions during reasoning, mimicking human-like cognitive processes.
>
> - **Full Open Solution with Data and Training**
>
>    While methods like SketchPad and Refocus focus on tool-use capacity on advanced commercial MLLMs (such as GPT-4v/o), VGR provides a complete, open-source pipeline including data construction and training. VGR's end-to-end data construction pipeline provides the community with a followable, efficient data iteration framework. VGR also employs joint training of detection loss and autoregressive loss, enhancing the model's grounding capabilities. This allows VGR to enable visual reasoning ability for an open-source VLM with no inherent visual capabilities, while SketchPad and Refocus heavily rely on existing capabilities of closed-source MLLMs.
>
> - **Efficient Visual Memory Architecture**
>
>   VGR introduces a novel architecture for visual memory.  Unlike tool-based methods (such as Visual CoT, SketchPad, Refocus, you mentioned), that repeatedly process image crops, VGR pre-computes visual tokens once.  By using multi-level pooling, it allocates computation to the most critical regions efficiently.  As shown in our experiments, this design saves approximately 70% of tokens compared to baseline methods while achieving superior performance.
>
> >  W2: Further explanation on LLaVA’s any-resolution.
>
> We thank the reviewer for highlighting the design choice regarding LLaVA’s any-resolution encoding. Here we provide additional clarification on our motivation and trade-offs.
>
> First, LLaVA-Next originally adopts a dynamic high-resolution strategy. Our method extends this design by adding more templates. Since we apply a relatively large 4×4 pooling when downsampling visual tokens, we record the horizontal and vertical template divisions to accurately reassemble sub-image features in the image memory. This enables us to directly apply a crop-like tensor operation using the predicted bounding boxes.
>
> Then, it is worth noting that while we use 4×4 downsampling for sub-image features, we use 2×2 downsampling for replay image features because replay regions are more focused on question-relevant local areas and will benefit from higher visual resolution. Compared to directly cropping the image and re-encoding it through the ViT, our approach avoids an additional ViT forward process, reducing computation during both training and inference. Moreover, the entire process is implemented at the data-processing level and remains fully modeling architecture-agnostic.
>
> Finally, as reported in Table 7 (rows 3 and 4), the crop-based approach and the visual-memory-based approach achieve comparable performance on downstream tasks, while the memory approach shows slight advantages. Crop also introduces additional ViT computation and increases the number of visual tokens fed into the LLM. Therefore, given the similar task performance, we choose the more computationally efficient memory approach, which also aligns better with the original dynamic-resolution mechanism in LLaVA-Next.

---

> ### Author Response · Authors · 2025-11-25
> **Response to Reviewer joFM(2/3)**
>
> >   W3: Clarify the scope of reasoning in the visually grounded reasoning.
>
> We appreciate your notice on the difference of VGR with existing reasoning studies. We would like to discuss it and add this interesting discussion to the paper. Indeed, our work is inspired by existing work with RL for text reasoning (e.g., DeepSeek-R1). However, text might be a good vessel for logic, but it is not efficient to express visual components (color, patterns, high-frequency signals). Therefore, the core motivation of our work is to propose the use of the visual memory, a new efficient tool to store and reuse the visual signal, and combine it with the reasoning chain inspired by DeepSeek-R1 and other pioneer works. Because of this reason, our work addresses different things, e.g., how to design architecture that enables visual memory, and how to construct data and training to enable visual thinking in MLLM, instead of improving the reasoning result with RL.
>
> We noticed, as mentioned by DeepSeek-R1, reinforcement learning is not the only way to enable reasoning; supervised finetuning is particularly efficient for smaller models (the performance for distilled models outperforms RL models). We adopt this way to synthesize the visual reasoning chain to train the model. Our results suggest the same phenomenon: supervised finetuning is able to activate the complex reasoning ability of the model. We agree that RL is a very promising path to further boost the model's ability. But we also believe the method proposed by VGR is self-contained and effective. We hope it can become a good baseline or cornerstone for future studies.
>
> >   W4: Replay mechanism.
>
> We thank the reviewer for the insightful questions regarding the replay mechanism. We address the concerns from three perspectives:
>
> **On the annotation of replay regions:** As described in Section 4.2 of the main paper, we intentionally enlarge the bounding boxes to avoid extreme aspect ratios and to reduce the risk of truncating question-relevant foreground objects. Covering most of the relevant foreground is sufficient, and a strict 1:1 alignment is not necessary. Moreover, these bounding boxes are generated by a stronger annotator model rather than the original ground-truth annotations. During training, the giou loss converges to around 0.55.
>
> **On how inaccurate replay affects reasoning:** To directly examine this, we evaluate two inference settings:
>
>  (1) attaching a random box image tensor while keeping the text box token correct;
>
>  (2) replacing both the text box token and the image tensor with a random box.
>
>  The corresponding results are shown in the table below:
>
>  method| Box Text Token               |         replay image token               | V* Bench | DocVQA | MMStar | ChartQA | TextVQA | AI2D | RWQA
> ------------------|-------------------------|------|----------|-------------|--------|---------|---------|------|------
> VGR-7B | random     | random   |   30.6    | 49.2        |  34.2   |  56.3 |   49.1 | 60.7 |    42.6
> VGR-7B  | right     | random  |  33.5   |     50.1       |  34.9   |  58.1  |  48.5  | 61.3 |  43.1
>
> We observe substantial performance degradation in both cases. Incorrect visual features cause large negative effects even when the text token is correct, and performance drops further when both text and visual features are incorrect. This demonstrates that accurate replay predictions play a crucial role in reasoning.
>
> It's noted that in the first line, it does not mean that both the box token and the corresponding image feature are random. Instead, it means that we first randomly crop an image tensor with a box. At the same time, replace the text token of the correct box in the current output ID with the text token corresponding to this box.
>
> **On the relationship between replay and grounding:** Using lmms-eval, we evaluate LLaVA-Next and VGR on Referring Expression Comprehension(REC) of RefCOCO, RefCOCOg, and RefCOCO+, reporting both the standard IoU>0.5 accuracy and the IoU metric. Results are shown below:
>
> Method                 | refcoco_acc@0.5 | refcoco_iou| refcocog_acc@0.5 | refcocog_iou| refcoco+_acc@0.5 |refcoco+_iou
> ----------------|----------------|------------------- |----------------|------------------- |--|--
> LLaVA-NeXT-7B            |     0.85  | 0.72 | 0.82        | 0.69 | 0.77|0.65
> VGR-7B            |  0.88    | 0.74  | 0.84      | 0.71| 0.78|0.68
>
>  The results show consistent improvements over the baseline, suggesting that VGR training and data also enhance grounding performance.

---

> ### Author Response · Authors · 2025-11-25
> **Response to Reviewer joFM(3/3)**
>
> >  W5:  Conduct VGR on other backbones.
>
> We appreciate the reviewer’s suggestion regarding evaluating additional backbones.  VGR is a unified framework involving both architectural design and data construction, so we require a baseline that can be fully trained from scratch with a reasonable implementation cost.  LLaVA-Next-7B fits this requirement well.
>
> In contrast, the training data for InternVL3 and Qwen2.5-VL is not publicly available.  Since VGR requires both pre-training and SFT, a complete reproduction of these data-closed models is not feasible.  Nevertheless, as reported in Table 3 of the main paper, we have already evaluated VGR on comparable baselines such as InternViT and Qwen2.5 LLM, where we still observe substantial improvements.
>
> To further address your concern, we conducted an additional analysis experiment.  We applied a post-training stage on InternVL3-8B using LLaVA-Next-770K data and obtained results comparable to those reported in its paper (although these data were already used during its internal training).  As a comparison, we then applied the VGR style post-training stage to InternVL3-8B under the same setting.  Note that this experiment includes only post-training rather than the full multi-stage VGR pipeline in our paper, and is therefore an analysis-oriented probe experiment. The results are shown below:
>
>  method| Data               |         Training&Inference Style               | DocVQA | MMStar | ChartQA | TextVQA | AI2D
> -----------|--------------------------------------------------------|------|----------|-------------|--------|---------|-------
> InternVL3-8B-posttrain | LLaVA-NeXT-770K     | General SFT   |   91.6    | 65.2       |  86.4   |  80.9 |  83.0
> InternVL3-8B-posttrain  | LLaVA-NeXT-770K+VGR-SFT     | VGR style  |  92.8   |     63.9       |  87.8   |  81.5  |  85.3
>
> Even without the full multi-stage pipeline, VGR-style post-training yields improvements on key benchmarks (DocVQA, ChartQA, TextVQA), indicating that the proposed strategy is transferable across different architectures.
>
> If you have any further questions or concerns, please feel free to contact us at any time. We are always available and looking forward to further discussions with you. :)
>
> Best regards,
>
> All Authors

---

> > ### Comment · Reviewer_joFM · 2025-11-28
> > **Official Comment by Reviewer joFM**
> >
> > I would like to thank the authors for their thorough and thoughtful rebuttal. After careful review, most of my initial concerns have been adequately addressed, and consequently, I will raise my score to 6.

---

> > > ### Author Response · Authors · 2025-11-28
> > >
> > > Thank you very much for your positive response.   We are very glad that most of your initial concerns have been addressed, and we sincerely appreciate your decision to raise the score.
> > >
> > > We are grateful for your thoughtful feedback and for engaging with our responses throughout the rebuttal process.
> > > During the remaining time of rebuttal, we still welcome you to have more discussions with us at any time
> > >
> > > Thank you again for your time and consideration.
> > >
> > > Best regards,
> > >
> > > All Authors

---

### Meta-Review · Area_Chair_T1DX · 2026-01-07

**Summary:**

Reviewer's major concerns are resolved and AC would like to make clear acceptance to this paper. We hope that the authors are not affected by this unconventional reviewing process.

**AC Note**

There are some irrelevant discussions among one of the reviews. When assessing the paper, AC decided to skip all relevant discussions. As an unbiased adjustment, the related comments/review/response from authors, reviewers, and all public comments will be skipped. AC thanks the original AC to provide clear guidance for carrying on the reviewing process.

The decision simply comes from the fact that these discussions are too noisy for the paper reviewing process, and we want to provide fair decisions for all papers. AC apologize on the time efforts that reviewers and authors have spent. AC's overriding does not reflect that the original review has unprofessional behavior. Meanwhile, AC is strongly against any personal attack and emotional statements in the ICLR reviewing process.

**Reviewer Concerns:**

Remaining concerns:

1. **The title "visually grounded reasoning"'s scope is under-specified** (joFM). I think that it is a good question but is also a minor one.
2. **Only 7B models are experimented; not larger one** (CQgV). The concern from the reviewers is valid. The scalability is a critical criterion for proposed method. It affects the paper's potential impact but we would not force the authors to go beyond their compute resource.

Resolved concerns:
1. **Lack of Technical Contribution** (joFM, qwzK). The reviewer is satisfied with author's clarification.
2. **Clarity of using any-resolution encoding instead of multi-crops** (joFM). Resolved by author's explanation in response.
3. **Missing analysis on replay mechanism** (joFM). Resolved by additional results.
4. **Whether VGR works on other backbone** (joFM). Resolved by additional results.
5. **Unclear in-domain and out-of-domain benchmarks** (CQgV). This is a paper presentation issue and the author claimed that they will solve it in final version.
6. **Missing computational cost and latency** (CQgV, ysjS). Resolved during rebuttal.
7. **The robustness of the replay** (ysjS). Answered point-by-point by the authors with clarification and additional results.
8. **Comparisons to other zoom-then-behavior baselines.** (ysjS). Additional results provided by the author.
8. **Lack of qualitative analysis** (qwzK). Additional results provided by the author.

**Reviewer Scores:**

Updated score: **6 6 6 8**

Original score: 4 4 6 6 8

**Details**
1. Two reviewers provided the score of 4 in original review. As most of the questions are resolved during rebuttal, they are willing to raise their score to 6. This action is reflected in their official public comments and discussions.
2. There are a bunch of irrelevant discussions among one specific review, which makes the paper assessment to be more noisy by considering such discussions. Thus the paper decision process exclude the review and all related discussions.

---

### Decision · Program_Chairs · 2026-01-26

Accept (Poster)